# FedPrecise: Stage-Precise Diffusion Descriptors for One-Shot Federated Learning

## Abstract

One-shot Federated Learning (OSFL) methods based on pretrained Latent Diffusion Models (LDM) have recently gained increasing attention due to their remarkable effectiveness in synthesizing high-fidelity and diverse training data on the server from client-side signals. However, uniform conditioning with a single client embedding used by previous approaches overlooks diffusion's stage-wise structure, in which different timestep stages govern corresponding semantic and frequency attributes, thereby blurring stage-specific styles, inducing distribution shifts in the synthesized data, this in turn degrades downstream classification performance, especially in highly heterogeneous domains such as medical imaging. To address this challenge, we propose FedPrecise, a stage-precise OSFL framework that personalizes the pretrained LDM at the diffusion-stage level via compact token descriptors. FedPrecise learns a set of stage-precise tokens on each client that modulate the pretrained LDM differently across denoising stages, enabling the server to generate synthetic images that more precisely reconstruct client distributions without transmitting auxiliary data beyond lightweight token descriptors. FedPrecise is also the first approach that explicitly addresses feature space heterogeneity using diffusion's stage-wise generation structure in OSFL. Extensive experiments on standard OSFL benchmarks with feature space heterogeneity show that FedPrecise consistently outperforms representative diffusion-based OSFL baselines in classification accuracy, including on challenging medical and satellite image datasets, while achieving substantially lower communication overhead than all baselines.

[1]Anonymous Institution, Anonymous City, Anonymous Region, Anonymous Country. Correspondence to: Anonymous Author <anon.email@domain.com>.

Preliminary work. Under review by the International Conference on Machine Learning (ICML). Do not distribute.

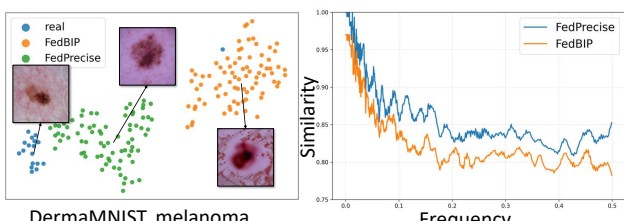

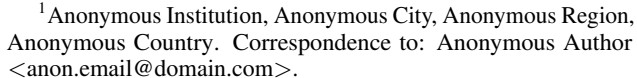

*Figure 1.* Feature space visualization of client images from the melanoma category in the DermaMNIST dataset, along with the generated images from FedBiP and FedPrecise. Additionally, the frequency-angular similarity curves of the two generated datasets and the real client images are shown. FedPrecise consistently exhibits higher angular similarity across a broad range of frequencies, indicating that the images generated by FedPrecise are closer to the original client images (Celentano & Di Lecce, 1997).

## 1. Introduction

Federated learning (FL) (McMahan et al., 2017) enables decentralized and privacy-preserving collaborative training of neural networks by keeping raw data on local clients while coordinating model updates through a central server. However, classical multi-round FL requires frequent bidirectional communication between clients and the server, which can incur substantial communication overhead (Kairouz et al., 2021). Moreover, frequently exchanging model updates exposes additional attack surfaces, making FL systems potentially vulnerable to adversarial behaviors such as Membership Inference Attacks (Shokri et al., 2017).

To mitigate these limitations, One-Shot Federated Learning (OSFL) (Guha et al., 2019) restricts collaboration to a single round of client–server communication. By allowing each client to upload only once, OSFL substantially improves communication efficiency. Moreover, fewer transmissions reduce the risk of data interception and the opportunities for adversarial attacks (Mendieta et al., 2025). Nevertheless, prior OSFL approaches (Guha et al., 2019; Li et al., 2020a; Lin et al., 2020) rely on auxiliary public datasets, which are difficult to acquire in privacy-sensitive domains such as healthcare (Liang et al., 2025; Liu et al., 2021). Some require clients to transmit richer information such as model parameters (Zhang et al., 2022), or distilled data (Zhou et al., 2020), thereby increasing communication cost and potentially elevating privacy risks. Moreover, these works

underemphasize the challenge of feature space heterogeneity across client data distributions (Chen et al., 2023) as well as the strict limitation on the quantity of local data (Wang et al., 2023; McMahan et al., 2017), which can be even more severe in rare domains (So et al., 2022).

Recently, diffusion models (Ho et al., 2020), particularly Latent Diffusion Models (LDMs) (Rombach et al., 2022), have attracted substantial attention due to their strong capability to synthesize high-quality images and their flexibility for controlled generation. In the OSFL setting, existing studies have shown that a pretrained LDM can be leveraged as a powerful server-side generator via a conditional description (Yang et al., 2024), and in some cases additional data-derived latent information (Chen et al., 2025), enabling the server to synthesize client-like datasets for training an effective global model. However, generating a single textual embedding across all diffusion steps limits the ability for visual attribute disentanglement (Alaluf et al., 2023). Prior work (Zhang et al., 2023) demonstrates that diffusion models generate images in the order of layout → content → material/style. Further analysis reveals that this generation order is correlated with the signal frequency of the corresponding attributes, progressing from low to high (Yang et al., 2023b; Lee et al., 2025). Thus, uniform conditioning overlooks this stage-wise structure, blurring stage-specific styles and thereby inducing distribution shifts in the synthesized data, which is particularly evident in highly feature space heterogeneous OSFL.

Therefore, in this paper, we propose Stage-Precise Diffusion Descriptors for One-Shot Federated Learning (FedPrecise) to address feature space heterogeneity and generate high-quality synthetic datasets that closely match client distributions. FedPrecise leverages and uploads only the locally learned stage-wise descriptors together with an adaptive segmentation mechanism to guide the server-side LDM in a multi-stage generation manner, precisely capturing visual attributes across clients. As shown in Figure 1, FedPrecise synthesizes feature-precise training data that more faithfully reconstructs client distributions, and consequently produces images whose angular similarity is closer to that of real samples across different signal-frequency ranges, thereby improving the performance of the downstream global classification model trained on the generated dataset without compromising communication efficiency. The main contributions of this work are summarized as follows:

- We propose FedPrecise, a novel OSFL framework for heterogeneous clients that is the first to leverage diffusion's stage-wise generation structure to personalize a pretrained LDM for mitigating feature space heterogeneity under a one-shot communication budget.

- We evaluate FedPrecise on three OSFL benchmarks with feature space heterogeneity and show that it de-

livers consistently strong performance, outperforming representative diffusion-based OSFL baselines in downstream classification accuracy.

- We further validate FedPrecise on two label space heterogeneous datasets from medical and satellite imaging, and conduct a detailed communication cost analysis, demonstrating that FedPrecise achieves favorable accuracy-communication trade-offs compared with prior OSFL methods.

## 2. Background and Related Work

### 2.1. One-Shot Federated Learning

Conventional federated learning (McMahan et al., 2017; Li et al., 2020b) typically requires multiple communication rounds. To reduce communication overhead and lower sensitivity to adversarial risks, existing one-shot federated learning (OSFL) methods mainly follow three directions. One relies on public auxiliary datasets to facilitate model distillation (Guha et al., 2019) or the transfer of client-side knowledge (Li et al., 2020a; Lin et al., 2020). Another focuses on aggregation of locally trained models, leveraging strategies such as loss surface adaptation (Su et al., 2023a) and Bayesian methods (Hasan et al., 2024; Yurochkin et al., 2019). These approaches may underutilize the rich distributional information available at the clients. A further line of work instead transfers data from clients to the server. This includes distilled data representations, data distributions, as well as approaches based on VAEs and GANs. With the success of Latent Diffusion Models (LDMs), recent OSFL methods have demonstrated that transmitting local data descriptors (Yang et al., 2024), combined with latent information through bi-level personalization (Chen et al., 2025), or by training local classifiers to guide the generation process can achieve superior global training performance compared to the aforementioned two directions (Yang et al., 2025). However, for the reasons discussed in Section 1, these methods still suffer from distribution shift, making it difficult to reconstruct synthetic datasets that faithfully reflect client data distributions.

### 2.2. Diffusion Models with Stage-wise Structure

Diffusion models (Ho et al., 2020; Song et al., 2021) have recently attracted growing attention for their strong capability of generating high-resolution images. In particular, Latent Diffusion Models (LDMs) (Rombach et al., 2022) have been widely adopted in various conditional generation applications due to their strong controllability and editability. By conditioning on text (Kim et al., 2022; Preechakul et al., 2022), images (Saharia et al., 2022; Su et al., 2023b), or other information, LDMs can synthesize diverse real-world images and support personalization via conditioning

*Figure 2.* Visualization of the difference between the stage-wise conditioning adopted by FedPrecise and the standard uniform conditioning. FedPrecise uses multi-stage tokens that respectively capture the client dataset's layout $\rightarrow$ content $\rightarrow$ material/style attributes, thereby enabling a more precise reconstruction of $D_{\text{syn}}$.

methods. Recent analyses further suggest that the denoising procedure of diffusion models exhibits a noteworthy stage-wise structure, where different timestep stages govern corresponding semantic attributes. This process can be characterized as progressing from layout to content and then to material/style attributes (Zhang et al., 2023). Complementary evidence indicates that the signal frequency associated with these attributes evolves from low to high along the generation process (Yang et al., 2023b; Lee et al., 2025; Choi et al., 2022). Motivated by these observations, prior works (Kynkäänniemi et al., 2024; Hang et al., 2023; Mahajan et al., 2024) have advocated applying dependent timestep conditioning or guidance strategies across different denoising stages, rather than using a uniform conditioning signal throughout the entire process, and have reported improved generation quality and controllability.

## 3. Method

### 3.1. Preliminaries

**Notations and Objectives.** We consider an OSFL setting with $K$ clients. Client $k$ has dataset $D_k = \{(x_i^{(k)}, y_i^{(k)})\}_{i=1}^{n_k}$. Let $f_\phi$ be a global classifier with parameters $\phi$. The ideal objective is:

$$\min_{\phi} \ F(\phi) = \frac{1}{K} \sum_{k=1}^{K} \mathbb{E}_{(x,y)\sim D_k}\big[\ell(f_\phi(x), y)\big], \quad (1)$$

where $\ell$ is the CE loss. In FedPrecise, the server optimizes this objective approximately by training on a synthetic dataset $D_{\text{syn}}$ generated with uploaded client descriptions:

$$\min_{\phi} \ \mathbb{E}_{(x,y)\sim D_{\text{syn}}}\big[\ell(f_\phi(x), y)\big]. \quad (2)$$

**Pre-trained latent diffusion model.** FedPrecise uses a fixed latent diffusion model (LDM) shared by all clients and the server. The LDM consists of an encoder $\varepsilon$, decoder $\mathcal{D}$, and denoising UNet $\epsilon_\theta$. Given an image $x_0$, the encoder produces a latent:

$$z_0 = \varepsilon(x_0). \quad (3)$$

A forward noising process generates $z_t$ at timestep $t \in \{0, \ldots, T\}$:

$$z_t = \sqrt{\alpha_t}\, z_0 + \sqrt{1 - \alpha_t}\, \epsilon, \quad \epsilon \sim \mathcal{N}(0, I). \quad (4)$$

Here $\{\alpha_t\}_{t=0}^{T}$ is a predefined noise schedule. Conditioned on an embedding $e$, the UNet predicts the corresponding noise term $\hat{\epsilon} = \epsilon_\theta(z_t, t \mid e)$, which is then used by the DDPM sampler to propagate the reverse denoising dynamics and obtain a final latent at $t = 0$. The noise prediction network $\epsilon_\theta$ is optimized by minimizing the following standard objective (Rombach et al., 2022):

$$\mathcal{L}_{\text{LDM}} = \mathbb{E}_{\varepsilon(x),\, \epsilon\sim\mathcal{N}(0,I),\, t}\left[\|\epsilon - \epsilon_\theta(z_t, t)\|_2^2\right]. \quad (5)$$

### 3.2. Local Adaptive Segmentation

It is observed that diffusion models behave differently across timesteps, motivating the use of stage-wise conditioning shown in Figure 2. Moreover, these temporal patterns further vary across client data distributions, which calls for a mechanism that can adapt to such variation. Consequently, FedPrecise exploits this by deriving a data-driven segmentation of the diffusion timesteps. A schematic overview of the FedPrecise framework is illustrated in Figure 3.

**Local denoising-difficulty curve.** For each timestep $t$, client $k$ characterizes its local denoising difficulty by measuring the noise-prediction MSE following Eq. (5) under a fixed base embedding $e_{\text{base}}$ as shown in the appendix:

$$\ell_k(t) = \mathbb{E}_{x_0\sim D_k,\, \epsilon\sim\mathcal{N}(0,1)}\left[\big\|\epsilon_\theta(z_t, t \mid e_{\text{base}}) - \epsilon\big\|_2^2\right], \quad (6)$$

where $z_t$ is obtained via Eq. (4). Directly computing $\ell_k(t)$ over the entire client dataset is unnecessary and computationally expensive. Instead, we approximate $\ell_k(t)$ with a Monte-Carlo estimator $\widehat{\ell}_k(t)$ based on $M$ stochastic evaluations and each freshly sampled Gaussian noise. Under independent sampling, $\widehat{\ell}_k(t)$ provides an unbiased estimate of the expectation in Eq. 6:

$$\widehat{\ell}_k(t) = \frac{1}{M} \sum_{i=1}^{M} \big\|\epsilon_\theta\big(z_t^{(i)}, t \mid e_{\text{base}}\big) - \epsilon^{(i)}\big\|_2^2. \quad (7)$$

Here each summand corresponds to one stochastic evaluation: $(x^{(i)}, y^{(i)})$ is sampled from $D_k$, $\epsilon^{(i)} \sim \mathcal{N}(0, 1)$ is independently drawn Gaussian noise, and $z_t^{(i)}$ is obtained

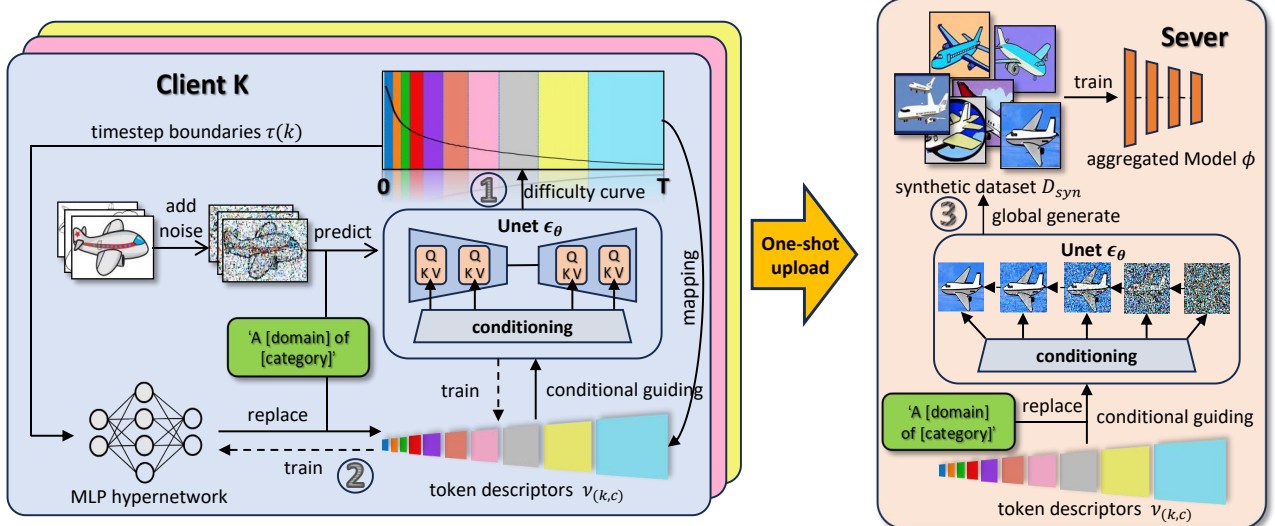

*Figure 3.* Overview of the FedPrecise framework. (1) Each client estimates a denoising-difficulty curve via a single forward diffusion process, from which a client-specific segmentation $\tau_k(t)$ is derived and aligned with local stage-wise conditions. (2) For each category, each client performs multiple update steps to train an MLP hypernetwork $\omega$, which is used to generate multi-stage descriptors $\mathcal{V}_{k,c}$. (3) The server collects the uploaded descriptors and segmentation information, synthesizes datasets using the pretrained LDM, and trains a global classifier $\phi$ on the resulting synthetic data.

by applying Eq. 4 to the corresponding input $x^{(i)}$. Finally, to reduce computation and suppress sampling noise, we uniformly divide $\{0, \ldots, T\}$ into $m$ blocks and evaluate the difficulty only at block representatives.

**Segmentation and stage mapping.** Given the difficulty curve $\widehat{\ell}_k[1{:}m]$, client $k$ partitions the diffusion timeline into $S$ contiguous stages with timestep boundaries:

$$\tau_k = (\tau_{k,0}, \tau_{k,1}, \ldots, \tau_{k,S}), \quad \tau_{k,0} = 0, \quad \tau_{k,S} = T. \quad (8)$$

We realize $\tau_k$ via block cut points $\{b_s\}_{s=0}^{S}$ by solving:

$$\min \sum_{s=1}^{S} \text{cost}(b_{s-1}+1, b_s) \quad \text{s.t.} \quad b_s - b_{s-1} \geq \Delta_{\min}, \quad (9)$$

where $\Delta_{\min}$ denotes the minimum block length, determined by the DDPM sampling interval used for image generation. For any block index range $[i, j]$, we first define the segment mean:

$$\text{cost}(i, j) = \sum_{p=i}^{j} \big(\widehat{\ell}_k[p] - \mu_{i,j}\big)^2, \quad (10)$$

where $\mu_{i,j}$ denotes the mean of $\widehat{\ell}_k[p]$ over range $[i, j]$, each timestep is finally assigned to a client-specific stage via:

$$\tau_k(t) = s \quad \text{iff} \quad \tau_{k,s-1} \leq t < \tau_{k,s}. \quad (11)$$

### 3.3. Local Descriptor Training

Given the client-specific segmentation $\tau_k(t)$, each client learns a set of stage-wise token descriptors that modulate

the diffusion process across different timesteps. For every class $c$ owned by client $k$, we parameterize a descriptor set:

$$\mathcal{V}_{k,c} = \{\, \mathbf{v}_{k,c}^{(1)}, \ldots, \mathbf{v}_{k,c}^{(S)} \,\}, \qquad \mathbf{v}_{k,c}^{(s)} \in \mathbb{R}^{d_w}, \quad (12)$$

where $d_w$ denotes the token embedding dimension of the textual conditioning model. We construct the conditional text embedding through a lightweight MLP hypernetwork, which generates the stage-wise token $\mathbf{v}_{k,c}^{(s)}$ and replaces it in the base prompt *"a [domain] of [category]"* at the *"[category]"* slot. For instance, this yields prompts such as *"a clipart style of a $[V_y]$"* on the DomainNet dataset. Formally, we denote the resulting descriptor embedding as $e_{\omega,k,c}^{(s)}$. The hypernetwork parameters $\omega$ are optimized locally on each client.

During local training, client $k$ samples a clean latent $z_0$ as defined in Eq. (3), selects a timestep $t$, and computes its stage index $s = \tau_k(t)$. A noisy latent $z_t$ is then generated by the standard forward process in Eq. (4). Following Eq. (5), the stage-wise tokens are optimized by minimizing the noise-prediction MSE:

$$\mathcal{L}_k = \mathbb{E}_{x_0 \sim D_k, \epsilon, t} \Big[ \big\| \epsilon_\theta\big(z_t, t \mid e_{\omega,k,c}^{(\tau_k(t))}\big) - \epsilon \big\|_2^2 \Big]. \quad (13)$$

Here only the hypernetwork parameters $\omega$ (and thus $\{\mathbf{v}_{k,c}^{(s)}\}$) are optimized, while the shared diffusion backbone remains frozen. This objective encourages each stage token to specialize to its corresponding temporal regime, while allowing cross-stage information sharing through the shared network. Notably, during the one-shot upload, each client only needs

to transmit extremely lightweight information $\{\mathbf{v}_{k,c}^{(s)}\}$ and $\tau_k(t)$ rather than MLP parameters $\omega$.

### 3.4. Global Generation and Training

**Image Generation.** After local training, the server holds the shared latent diffusion model and, for each client $k$ and class $c$, a stage-wise descriptor set $\{\mathbf{v}_{k,c}^{(s)}\}_{s=1}^{S}$ together with the stage mapping $\tau_k(t)$. To synthesize an image of class $c$ for client $k$, the server samples an initial latent:

$$z_T \sim \mathcal{N}(0, 1), \tag{14}$$

then integrates descriptors into the same pretrained LDM and generates synthetic images with:

$$\tilde{x}_i = \mathcal{D}\Big(\epsilon_\theta\big(z_T, T, e(\{\mathbf{v}_{k,c}^{(s)}\}_{s=1}^{S})\big)\Big), \tag{15}$$

where $\mathcal{D}(\cdot)$ denotes the decoder of LDM that reconstructs the image from the latent. At each timestep $t$, the sampler sets $s = \tau_k(t)$ and constructs the stage-wise descriptor embedding via $e(\mathbf{v}_{k,c}^{(s)})$ and $e(\cdot)$ replaces the given token into the base prompt and returns the corresponding text embedding.

In addition, we adopt classifier-free guidance (CFG) (Ho & Salimans, 2022) during image synthesis, which enables us to naturally incorporate text embeddings into the generation process while maintaining flexible control over conditional guidance at different denoising stages. After the generation process is completed, the data sample $\tilde{x}_i$ and its label $y_i$ are appended to the synthetic dataset $D_{\text{syn}}$ as:

$$D_{\text{syn}} = \{(\tilde{x}_i, y_i)\}. \tag{16}$$

**Aggregated Model Training.** Given $D_{\text{syn}}$, the server trains a global classifier $f_\phi$ (e.g., a ResNet-18) by minimizing the cross-entropy loss Eq. (2). All client-specific information here is conveyed through the stage-wise descriptors used during generation, so the resulting classifier meets the one-shot upload requirement in OSFL without accessing any raw client data.

## 4. Experiments and Analyses

### 4.1. Benchmark Experiments

**Dataset settings.** We evaluate FedPrecise on three large-scale real-world image benchmarks with feature distribution shift: *DomainNet* (Peng et al., 2019), *PACS* (Li et al., 2017), and *OfficeHome* (Venkateswara et al., 2017). These benchmarks span diverse visual styles and acquisition conditions, thereby inducing shifts representative of practical OSFL deployments. DomainNet contains six domains, including Clipart (C), Infograph (I), Painting (P), Quickdraw (Q), Real (R), and Sketch (S). Following FedBiP (Chen et al., 2025),

we select 10 categories within each DomainNet domain to construct the benchmark. PACS comprises four domains: Art (A), Cartoon (C), Photo (P), and Sketch (S), with 7 categories in each domain. OfficeHome contains four real-life object domains, including Art (A), Clipart (C), Product (P), and Real (R). For each domain, we consider 20 categories. To simulate the feature space heterogeneous OSFL scenario, each client is associated with a specific domain, yielding substantially different local data distributions across clients. We additionally consider the constraint of limited local data by adopting a few-shots setting: 24 shots per class for DomainNet and PACS, and 8 shots per class for OfficeHome.

**Baseline methods settings.** We compare FedPrecise with several baseline methods spanning centralized training, classical multi-round FL, and diffusion-based OSFL. **(1) Central** is an *oracle* method: it directly accesses and aggregates all clients' raw data, thereby fully violating client privacy, and serves as a reference ceiling for downstream classification performance. **(2) FedAvg** requires multi-round communication between clients and the server, and thus does not satisfy the one-shot communication requirement of OSFL. **(3) FENS** (Allouah et al., 2024) exhibits characteristics of both OSFL and FL. Besides, we evaluate diffusion-based OSFL baselines including **(4) FedDEO** (Yang et al., 2024), which optimizes a single full-sentence description on client and uploads it to condition a LDM for data synthesis. **(5) FedBiP** (Chen et al., 2025), which uploads additional latent representations to enable both instance-level and concept-level personalization. **(6) FedLMG** (Yang et al., 2025), which trains a classifier on each client and uploads the model parameters to supervise a diffusion model for generation. The implementation details of FedPrecise and additional details are provided in the **appendix**.

**Results and Analyses.** The benchmark results are reported in Table 1, and the visualization experiments are provided in Figure 6. We observe that FedPrecise achieves the best average performance among all baselines, with an average improvement of up to 3.36%. Notably, FedPrecise attains these gains while being guided only by 10 token descriptors, outperforming OSFL methods that upload local model parameters or additional information, as well as classical FL baselines that require multi-round communication. The advantage is particularly evident on domains with rich multi-stage visual attributes, such as Painting across all three benchmarks, Infograph in DomainNet, and Art in PACS. However, FedPrecise slightly underperforms on a few domains, e.g., Quickdraw in DomainNet and Sketch in PACS, which we attribute to the relatively homogeneous feature patterns in these domains, where stage-wise characteristics become more similar and thus limit the effectiveness of stage-precise guidance.

*Table 1.* Main results on DomainNet, PACS, and OfficeHome. Best and second best are highlighted in bold and underline, respectively (excluding Central, Prompt, and FedAvg).

| Dataset | | Central | Prompt | FedAvg | DENSE | FENS | FedLMG | FedDEO | FedBiP | FedPrecise |
|---|---|---|---|---|---|---|---|---|---|---|
| Domain Net | C | 85.63 | 81.99 | 81.99 | 73.95 | 77.97 | 75.29 | 78.93 | 82.11 | **87.36** |
| | I | 62.37 | 55.62 | 57.67 | 43.35 | 54.19 | 55.83 | 51.74 | 57.96 | **58.08** |
| | P | 79.69 | 71.65 | 73.56 | 63.99 | 69.20 | 67.31 | 71.01 | 71.25 | **72.80** |
| | Q | 86.36 | 39.09 | 43.64 | 42.91 | 69.15 | 57.88 | **70.79** | 67.89 | 69.42 |
| | R | 91.79 | 82.98 | 91.19 | 82.68 | 87.21 | 82.20 | 84.40 | 86.33 | **87.80** |
| | S | 85.50 | 69.72 | 79.52 | 72.52 | 75.57 | 73.92 | 74.56 | 70.22 | **81.81** |
| | **Avg** | 81.89 | 66.84 | 71.26 | 63.23 | 72.22 | 68.74 | 71.90 | 72.63 | **76.21** |
| PACS | A | 85.85 | 72.44 | 73.29 | 63.17 | 71.95 | 71.46 | 73.42 | 73.90 | **80.02** |
| | C | 84.86 | 72.28 | 78.35 | 57.14 | **82.30** | 75.91 | 79.96 | 79.53 | 80.74 |
| | P | 96.70 | 89.79 | 95.69 | 91.59 | 89.79 | 91.59 | 89.49 | 89.79 | **94.09** |
| | S | 80.53 | 56.87 | 57.16 | 47.96 | 58.78 | 59.29 | 65.27 | **74.68** | 71.48 |
| | **Avg** | 86.99 | 72.85 | 76.12 | 64.97 | 75.71 | 74.56 | 77.03 | 79.48 | **81.59** |
| Office Home | A | 61.44 | 43.79 | 52.84 | 32.03 | 49.02 | 35.29 | 43.79 | 50.63 | **50.98** |
| | C | 62.10 | 51.59 | 52.76 | 32.80 | 48.41 | 51.59 | **60.19** | 53.19 | 57.01 |
| | P | 79.62 | 69.81 | 71.80 | 53.96 | 72.83 | 60.76 | 71.32 | 72.83 | **78.49** |
| | R | 77.66 | 68.50 | 60.02 | 48.35 | 63.00 | 65.20 | 69.96 | 69.50 | **77.29** |
| | **Avg** | 70.20 | 58.42 | 59.36 | 41.79 | 58.32 | 53.21 | 61.32 | 61.54 | **65.94** |

## 4.2. Experiment on Real-World Medical and Satellite Image Datasets

To evaluate FedPrecise on challenging real-world data, we further conduct experiments on a medical dataset, *DermaMNIST* (Yang et al., 2023a), which contains dermatoscopic skin lesion images from seven categories, and a satellite-image dataset, the *UC Merced Land Use Dataset (UCMerced)* (Yang & Newsam, 2010), which includes 21 land use categories. To develop heterogeneous client datasets, we use a Dirichlet distribution to control the strength of label-space heterogeneity. A smaller Dirichlet concentration parameter $\alpha$ corresponds to a higher degree of data heterogeneity. Following this setting, we partition each dataset into five client datasets for evaluation.

The evaluation results are reported in Table 2. Across different levels of label-space heterogeneity, FedPrecise consistently outperforms all baselines, achieving an average improvement of 2.92 percentage points on DermaMNIST and 2.83 percentage points on UCMerced. This consistent gain across $\alpha$ suggests that the advantage of FedPrecise is not restricted to a particular regime, but persists under varying degrees of non-IIDness. As illustrated in Figure 4, the samples synthesized by FedPrecise markedly bridge the gap between the synthetic images produced by the current SOTA method FedBiP and the challenging real-world data, given the substantial distribution shift between medical/satellite images and common natural images , these results provide strong evidence that stage-wise personalization is effective for handling feature space heterogeneity, and experiments with varying heterogeneity levels controlled by $\alpha$ further demonstrate the competitive performance of FedPrecise under non-IID settings.

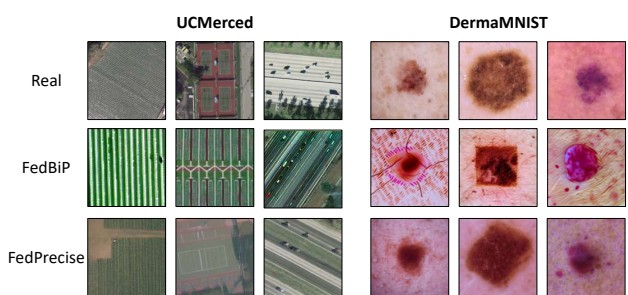

*Figure 4.* Comparison between FedBiP and FedPrecise on medical and satellite datasets. The images synthesized by FedPrecise are visibly closer to real samples, exhibiting more faithful background and material characteristics that better match the real features.

### 4.3. Ablation and Sensitivity Analyses.

**Ablation Study.** To further verify the necessity of each component in FedPrecise, we conduct an ablation study on the representative DomainNet benchmark, with results reported in Table 3. We first observe that using a prompt-based text embedding, *"a* [domain] *of* [category]*"*, to assist the generation of a synthetic dataset, a practice widely adopted by prior diffusion-based OSFL methods such as FedDEO, can slightly improve the performance of FedPrecise. Notably, even without prompt conditioning, FedPrecise still achieves a clear improvement over other baselines. We then find that employing *Local Adaptive Segmentation* brings a substantial performance gain compared with uniformly partitioning diffusion timesteps, which further supports that adaptive segmentation effectively collaborates with stage-wise conditioning to generate heterogeneous client-like data. Finally, under the same training budget, we compare training stage-wise descriptors with an *MLP hypernetwork* against

*Table 2.* Results on DermaMNIST and UCMerced with varying label-space heterogeneity ($\alpha$). Best and second best are highlighted in bold and underline.

| Dataset | $\alpha$ | FedAvg | Central | DENSE | FENS | FedLMG | FedDEO | FedBiP | FedPrecise |
|---|---|---|---|---|---|---|---|---|---|
| DermaMNIST | 5 | 61.34 | 68.274 | 54.469 | 59.399 | 36.125 | 56.459 | 58.155 | **62.045** |
|  | 0.5 | 53.74 | 68.274 | 45.434 | 55.147 | 27.905 | 46.783 | 53.167 | **57.656** |
|  | 0.01 | 46.75 | 68.274 | 39.733 | 29.177 | 26.272 | 43.741 | 47.631 | **51.222** |
| UCMerced | 5 | 77.631 | 85.857 | 66.284 | 66.185 | 50.952 | 78.095 | 79.286 | **81.905** |
|  | 0.5 | 74.913 | 85.857 | 60.584 | 62.843 | 44.476 | 74.286 | 76.019 | **78.571** |
|  | 0.01 | 70.022 | 85.857 | 51.985 | 32.618 | 30.942 | 69.048 | 71.429 | **74.762** |

*Table 3.* Ablation study for different components of FedPrecise. Best and second best are highlighted in bold and underline, respectively.

| MLP | 10-stage Seg | Prompt | Adaptive Seg | DomainNet |
|---|---|---|---|---|
| ✓ | ✓ | ✓ | ✓ | **76.043** |
| ✓ | ✓ | ✗ | ✓ | 75.241 |
| ✓ | ✓ | ✓ | ✗ | 73.334 |
| ✓ | ✓ | ✗ | ✗ | 73.288 |
| ✗ | ✓ | ✗ | ✗ | 69.252 |
| ✗ | ✗ | ✗ | ✗ | 66.361 |

*Table 4.* The influence of the number of diffusion stages.

| | DomainNet | | | | | | |
|---|---|---|---|---|---|---|---|
| *stages* | *C* | *I* | *P* | *Q* | *R* | *S* | *avg* |
| 10 | 87.356 | 58.078 | 72.797 | 68.424 | 87.798 | 81.807 | 76.043 |
| 5 | 83.908 | 56.033 | 71.392 | 65.212 | 85.357 | 79.644 | 73.591 |
| 3 | 84.100 | 61.350 | 71.137 | 57.636 | 86.429 | 78.244 | 73.149 |
| 1 | 81.801 | 61.350 | 73.436 | 55.273 | 82.857 | 77.481 | 72.033 |

both separately training ten descriptors registered as learnable parameters and directly optimizing a single token descriptor. The performance gap observed in the experimental results indicates that the descriptor optimization based on *MLP hypernetwork* is an indispensable component of FedPrecise, playing an important role in capturing shared foundational features across descriptors while enabling differentiated stage-specific adaptation.

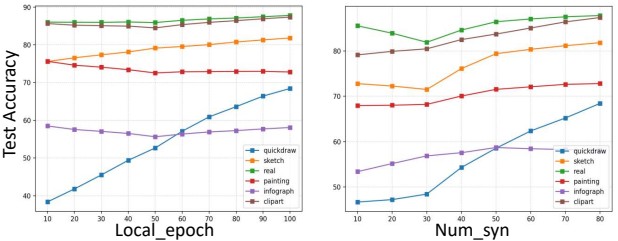

*Figure 5.* Accuracy curves of FedPrecise w.r.t. the number of local training epochs and the number of synthesized images per class.

**Sensitivity Analyses.** We further conduct sensitivity analyses on DomainNet with respect to local training epochs, the number of synthesized images per class in Figure 5, and the number of diffusion stages in Table 4. Overall, increasing either the local training epochs or the synthesis budget leads to consistent but moderate improvements, suggesting that FedPrecise benefits from stronger local fitting and richer synthetic data coverage. More importantly, the number of diffusion stages plays a pivotal role: using too few stages forces diverse visual attributes across denoising phases to be mixed within the same stage, which weakens the precision of stage-wise guidance and degrades downstream performance. In contrast, adopting excessively many stages may

increase communication overhead and also makes each descriptor govern an overly narrow denoising range, which can be harder to train sufficiently. As shown in Table 4, stage $= 10$ provides a strong and stable choice with favorable accuracy, which is also a well motivated prior following ProSpect (Zhang et al., 2023). Thus, our default setting is fully reasonable.

## 5. Communication Cost and Privacy Validation

**Communication Cost Validation.** We provide a theoretical communication-cost analysis of FedPrecise and representative baselines, as summarized in Table 5. We assume that each client contains 10 categories. For FL-style methods, the number of communication rounds is set to 50, and all methods adopt ResNet-18 as the classifier backbone, consistent with the settings in our benchmark experiments. The baselines can be broadly categorized into two communication patterns. *Uploading model parameters* (e.g., FedLMG, DENSE, and FENS) requires each client to transmit the locally trained classifier parameters, which induces a relatively large one-shot communication cost in OSFL. In contrast, *uploading descriptor information* (e.g., FedDEO, FedBiP, and FedPrecise) only transmits client-side descriptions to condition the server-side generator for obtaining a synthetic dataset, leading to smaller communication overhead under a one-shot budget. Notably, FedDEO uploads one full-sentence embedding per category, which consists of 77 vectors of 768 dimensions, while FedBiP further uploads additional latent representations for personalization. By comparison, FedPrecise only uploads stage-wise token descriptors, where each stage corresponds to a 768-dimensional token vector. This method yields the lowest

*Table 5.* Comparison of the communication costs among baseline methods.

| | | Uploaded Parameters (M) | | | | |
|---|---|---|---|---|---|---|
| *Avg* | *DEO* | *BiP* | *LMG* | *FENS* | *DENSE* | *Precise* |
| 5844.756 | 5.914 | 47.270 | 116.895 | 121.345 | 116.895 | **0.768** |

communication cost among all baselines, which becomes particularly advantageous when scaling to a large number of clients.

**Privacy Concerns.** Due to its one-shot upload setting, where potential privacy leakage mainly arises, FedPrecise incurs a lower privacy risk compared to multi-round FL methods in which information leakage may accumulate across repeated communications. We provide visualizations of privacy-sensitive and information-related categories in Figure 7 (e.g., faces and notebooks) to verify that it is difficult to obtain synthesized images containing specific client privacy information via FedPrecise. Such qualitative evidence complements quantitative attack evaluations by directly inspecting whether identifiable client-specific details are reproduced. We further evaluate FedPrecise under Membership Inference Attacks (MIA) following (Yeom et al., 2018), measuring the differences in the average loss and predictive entropy of the final model between training member data and non-member data. As shown in Table 6, FedPrecise exhibits smaller gaps in both loss and entropy compared to FedAvg and FedBiP, indicating stronger membership privacy preservation.

*Table 6.* MIA analysis on different datasets with FedAvg, FedBiP, and FedPrecise.

| Dataset | MIA Metric | FedAvg | FedBiP | FedPrecise |
|---|---|---|---|---|
| DomainNet | Entropy ↓ | 0.157 | 0.017 | 0.010 ↓93.9% |
| | Loss ↓ | 0.484 | 0.123 | 0.078 ↓84.0% |
| DermaMNIST | Entropy ↓ | 0.072 | 0.181 | 0.029 ↓60.0% |
| | Loss ↓ | 0.712 | 0.455 | 0.255 ↓64.2% |
| PACS | Entropy ↓ | 0.199 | 0.152 | 0.074 ↓62.7% |
| | Loss ↓ | 0.373 | 0.206 | 0.223 ↓40.1% |

# 6. Conclusions

In this work, we propose FedPrecise, the first OSFL framework that leverages diffusion models' stage-wise structure to address feature space heterogeneity. FedPrecise learns stage-precise descriptors together with an adaptive segmentation mechanism to capture client-specific visual attributes, thereby enabling the server to more precisely reconstruct client data distributions and alleviating feature shifts in the resulting synthetic dataset. Extensive experiments demonstrate the effectiveness of FedPrecise under both feature space and label space heterogeneous settings, with partic-

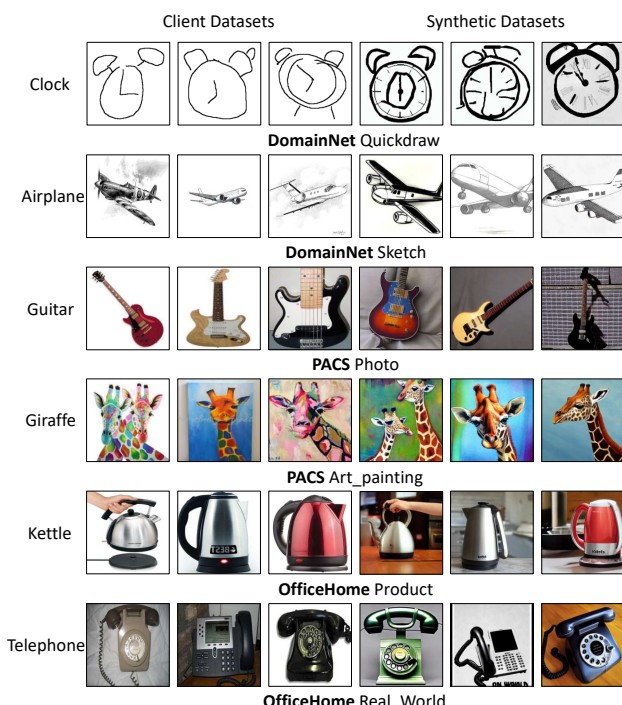

*Figure 6.* Visualization of generated samples on different datasets.

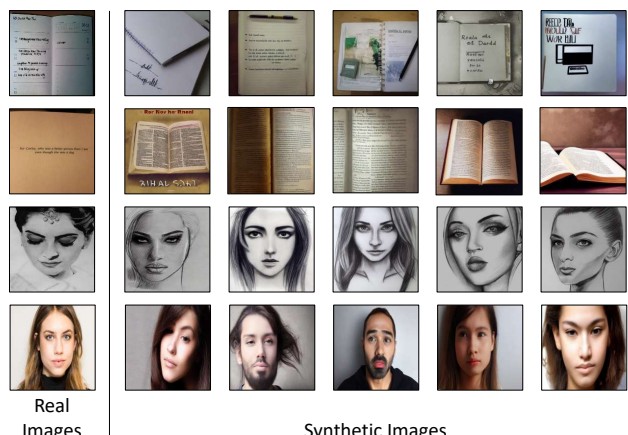

*Figure 7.* Visualization of privacy-sensitive information-related categories, where the generated images are obtained by retrieving the top-5 synthetic images most similar to the corresponding real images.

ularly strong performance on rare domains such as medical and satellite imagery. Our ablation study further confirms the necessity of adaptive segmentation and highlights the fundamental role of the MLP component in optimizing stage-wise descriptors. Finally, the communication-cost and privacy evaluations show that FedPrecise maintains privacy-preserving performance while offering clear advantages when scaling to a large number of clients, even compared with OSFL baselines that already have relatively low communication overhead.

## Impact Statement

This paper presents work whose goal is to advance the field of machine learning. There are many potential societal consequences of our work, none of which we feel must be specifically highlighted here.

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

## A. Proofs

In this section, we provide a proof of the theoretical analyses, regarding upper bound of the KL divergence between the client data distribution and the stage-wise conditional generation distribution of the diffusion model.

We denote the client data distribution by

$$p_n(x), \tag{17}$$

and the (unconditional) distribution induced by the diffusion model (DM) by

$$p_{\epsilon_\theta}(x). \tag{18}$$

We represent the stage-wise condition as a length-$K$ tuple

$$d^{1:K} \triangleq \left(d^{(1)}, d^{(2)}, \dots, d^{(K)}\right), \tag{19}$$

and the corresponding stage-wise conditional generation distribution as

$$p_{\epsilon_\theta}(x \mid d^{1:K}). \tag{20}$$

For convenience, we also denote the partial-conditioning distribution by

$$p_{\epsilon_\theta}(x \mid d^{1:k}), \tag{21}$$

$$p_{\epsilon_\theta}(x \mid d^{1:0}) \triangleq p_{\epsilon_\theta}(x). \tag{22}$$

We interpret

$$d^{1:K} \tag{23}$$

as a fixed condition sequence.

**Assumption A.1.**

$$\mathrm{KL}\big(p_n(x) \,\|\, p_{\epsilon_\theta}(x)\big) \leq \lambda. \tag{24}$$

This assumption posits that the unconditional diffusion model provides a distributional reference that is not arbitrarily far from the client distribution. Considering that the LDM used in FedPrecise has been sufficiently pretrained on data drawn from most common distributions, it can generate images that cover the client distribution; hence, this assumption is natural.

**Assumption A.2.**

$$\mathbb{E}_{x \sim p_n}\big[\log p_{\epsilon_\theta}(x \mid d^{1:k}) - \log p_{\epsilon_\theta}(x \mid d^{1:k-1})\big] \geq \gamma_k. \tag{25}$$

This assumption formalizes the fact that incorporating the $k$-th stage condition does not decrease the expected log-likelihood on client data. Since the stage-wise descriptors are sufficiently trained on the same underlying real data distribution, appending an additional descriptor on top of the previous-stage conditioning to guide one stage of the denoising process naturally makes the resulting image distribution more determined. Therefore, it is reasonable to assume a nonnegative lower bound $\gamma_k$ on the expected log-likelihood gain at each stage. Consequently, we obtain:

**Theorem A.3.**

$$\mathrm{KL}\big(p_n(x) \,\|\, p_{\epsilon_\theta}(x \mid d^{1:K})\big) \leq \lambda - \sum_{k=1}^{K} \gamma_k. \tag{26}$$

*Proof.* By the definition of KL divergence,

$$\mathrm{KL}\big(p_n(x) \,\|\, p_{\epsilon_\theta}(x \mid d^{1:K})\big) = \int p_n(x) \, \log \frac{p_n(x)}{p_{\epsilon_\theta}(x \mid d^{1:K})} \, dx. \tag{27}$$

Multiplying and dividing by

$$p_{\epsilon_\theta}(x)$$

inside the logarithm yields

$$\mathrm{KL}\big(p_n(x) \,\|\, p_{\epsilon_\theta}(x \mid d^{1:K})\big) = \mathrm{KL}\big(p_n(x) \,\|\, p_{\epsilon_\theta}(x)\big) + \mathbb{E}_{x \sim p_n}\big[\log p_{\epsilon_\theta}(x) - \log p_{\epsilon_\theta}(x \mid d^{1:K})\big]. \tag{28}$$

Applying Eq. (1) to Eq. (5) gives

$$\text{KL}\big(p_n(x) \,\|\, p_{\epsilon_\theta}(x \mid d^{1:K})\big) \leq \lambda + \mathbb{E}_{x \sim p_n}\big[\log p_{\epsilon_\theta}(x) - \log p_{\epsilon_\theta}(x \mid d^{1:K})\big]. \tag{29}$$

Next, we telescope over stages:

$$\log p_{\epsilon_\theta}(x) - \log p_{\epsilon_\theta}(x \mid d^{1:K}) = -\sum_{k=1}^{K}\Big[\log p_{\epsilon_\theta}(x \mid d^{1:k}) - \log p_{\epsilon_\theta}(x \mid d^{1:k-1})\Big]. \tag{30}$$

Taking expectation with respect to

$$x \sim p_n$$

yields

$$\mathbb{E}_{x \sim p_n}\big[\log p_{\epsilon_\theta}(x) - \log p_{\epsilon_\theta}(x \mid d^{1:K})\big] = -\sum_{k=1}^{K}\mathbb{E}_{x \sim p_n}\big[\log p_{\epsilon_\theta}(x \mid d^{1:k}) - \log p_{\epsilon_\theta}(x \mid d^{1:k-1})\big]. \tag{31}$$

Applying Eq. (2) term-wise to Eq. (8) gives

$$\mathbb{E}_{x \sim p_n}\big[\log p_{\epsilon_\theta}(x) - \log p_{\epsilon_\theta}(x \mid d^{1:K})\big] \leq -\sum_{k=1}^{K}\gamma_k. \tag{32}$$

Combining Eq. (6) and Eq. (9) completes the proof. $\qquad\square$

*Remark* A.4 (Mutual-information form and comparison to FedDEO). We next provide an information-theoretic rewriting for comparison. Here we consider the expected conditional KL under the data-induced joint distribution.

**Stage-wise (ours).** Define the joint distribution

$$q(x, d^{1:K}) \triangleq p_n(x)\, p_{\epsilon_\theta}\big(d^{1:K} \mid x\big), \tag{33}$$

and its marginal

$$q(d^{1:K}) \triangleq \int p_n(x)\, p_{\epsilon_\theta}\big(d^{1:K} \mid x\big)\, dx. \tag{34}$$

Also define the model-implied marginal of

$$d^{1:K} \tag{35}$$

under

$$p_{\epsilon_\theta}(x) \tag{36}$$

by

$$p_{\epsilon_\theta}(d^{1:K}) \triangleq \int p_{\epsilon_\theta}(x)\, p_{\epsilon_\theta}\big(d^{1:K} \mid x\big)\, dx. \tag{37}$$

The mutual information under

$$q \tag{38}$$

is

$$I\big(X; d^{1:K}\big) \triangleq \mathbb{E}_{(x, d^{1:K}) \sim q}\left[\log \frac{p_{\epsilon_\theta}\big(d^{1:K} \mid x\big)}{q(d^{1:K})}\right]. \tag{39}$$

A direct Bayes expansion yields the exact decomposition

$$\mathbb{E}_{d^{1:K} \sim q}\big[\text{KL}\big(p_n(x) \,\|\, p_{\epsilon_\theta}(x \mid d^{1:K})\big)\big] = \text{KL}\big(p_n(x) \,\|\, p_{\epsilon_\theta}(x)\big) - I\big(X; d^{1:K}\big) - \text{KL}\big(q(d^{1:K}) \,\|\, p_{\epsilon_\theta}(d^{1:K})\big). \tag{40}$$

Since the last term is nonnegative, Eq. (10) implies

$$\mathbb{E}_{d^{1:K} \sim q}\big[\text{KL}\big(p_n(x) \,\|\, p_{\epsilon_\theta}(x \mid d^{1:K})\big)\big] \leq \text{KL}\big(p_n(x) \,\|\, p_{\epsilon_\theta}(x)\big) - I\big(X; d^{1:K}\big), \tag{41}$$

and by Eq. (1),

$$\mathbb{E}_{d^{1:K} \sim q}\big[\text{KL}\big(p_n(x) \,\|\, p_{\epsilon_\theta}(x \mid d^{1:K})\big)\big] \leq \lambda - I\big(X; d^{1:K}\big). \tag{42}$$

**Single-stage (FedDEO).** FedDEO (Yang et al., 2024) proves the following single-stage upper bound (their Eq. (8)):

$$\text{KL}\big(p_n(x) \,\|\, p_{\epsilon_\theta}(x \mid d)\big) < \lambda + \mathbb{E}[\log p_{\epsilon_\theta}(d)] - \int p_n(x) \, \log p_{\epsilon_\theta}(d \mid x) \, dx. \tag{43}$$

For the same data-induced joint distribution defined by

$$q(x, d) \triangleq p_n(x) p_{\epsilon_\theta}(d \mid x), \tag{44}$$

with

$$q(d) \triangleq \int p_n(x) \, p_{\epsilon_\theta}(d \mid x) \, dx, \qquad p_{\epsilon_\theta}(d) \triangleq \int p_{\epsilon_\theta}(x) \, p_{\epsilon_\theta}(d \mid x) \, dx, \tag{45}$$

the mutual information under

$$q \tag{46}$$

is

$$I(X; d) \triangleq \mathbb{E}_{(x,d) \sim q} \left[ \log \frac{p_{\epsilon_\theta}(d \mid x)}{q(d)} \right]. \tag{47}$$

Dropping the nonnegative mismatch term and applying Eq. (1) yields

$$\mathbb{E}_{d \sim q} \big[ \text{KL}\big(p_n(x) \,\|\, p_{\epsilon_\theta}(x \mid d)\big) \big] \leq \lambda - I(X; d). \tag{48}$$

Considering that our remark derives two upper bounds and shows that the KL-divergence upper bound of FedDEO and the form of FedPrecise are unified, this unification reflects common distributional characteristics of diffusion-model generation. Moreover, in practice, since the multi-stage conditions typically carry stronger information than a single-stage condition, the resulting upper bound of FedPrecise is empirically tighter.

## B. Experimental Details

**FedPrecise implementation.** We conduct all experiments on 3 NVIDIA RTX 3090 GPUs (24GB memory each) and implement our method in PyTorch. For the LDM backbone, we use the HuggingFace open-sourced `stable-diffusion-v1-5`, and adopt a ResNet-18 (He et al., 2016) pretrained on ImageNet (Deng et al., 2009) as the initialization of the classifier. For FedPrecise, we perform client-side segmentation with 10-stage token descriptors following Prospect (Zhang et al., 2023). During generation, we use classifier-free guidance (CFG) with `guidance_scale_text`=3 and `guidance_scale_desc`=3.5, which is consistent with our FedDEO implementation. The minimum step interval $\Delta_{\min}$ is determined by the DDPM sampling interval used for image generation, and we set $\Delta_{\min} = 20$ for all experiments. For efficiency, we set $m = 100$.

Our hypernetwork is implemented as a gated MLP with residual preprocessing. Specifically, it refines a 768-dimensional input using two parallel low-rank linear adapters with bottleneck dimension 512, and then projects it to a 7680-dimensional output (10-stage) via a GEGLU-based feed-forward network with hidden dimension 3072. We use a learning rate of 0.001 for training the hypernetwork.

**Baselines methods.** For a fair comparison among OSFL methods including FedDEO, FedBiP, FedLMG, and FedPrecise, we use ResNet-18 as the global classifier for all baselines. We fix the number of synthesized images to 80 per class, set the local training epochs for client models and embeddings to 100, and set the epochs of the global training stage (OSFL) as well as the communication rounds (FL) to 50.

In particular, for FedAvg, we set the number of local training epochs to 5. For FENS, it trains models on clients, uploads them to the server using OSFL, and then adopts an FL-style training process on the server to learn a prediction aggregator model. For FedDEO, we register the descriptor as a trainable parameter of size $77 \times 768$ and optimize it locally. For FedBiP, it locally optimizes a domain concept vector together with a category concept vector. For FedLMG, we train a local classifier and upload it to the server to guide image generation. Finally, all methods that require textual prompts follow (Chen et al., 2025).

---

**Algorithm 1** Training process of FedPrecise

---

1: **GlobalUpdate**
2: Initialize LDM with pretrained weights $\theta$, Initialize classification model $\phi$, synthetic dataset $D_{\text{syn}}$
3: **for** client $k = 1$ to $K$ **in parallel do**
4:     $k$-th client executes ClientTrain($k$) and uploads stage boundaries $\tau_k = (\tau_{k,0}, \ldots, \tau_{k,S})$ and stage-wise descriptors
    $\{\mathbf{v}_{k,c}^{(s)}\}_{s=1}^{S}$ for each local class $c$
5: **end for**
6: **for** client $k = 1$ to $K$ **do**
7:     **for** each uploaded class $c$ of client $k$ **do**
8:         **for** $i = 1$ to $Num_{syn}$ **do**
9:             Sample $z_T \sim \mathcal{N}(0, I)$
10:             $\tilde{x}_i \leftarrow \mathcal{D}\Big(\epsilon_\theta\big(z_T, T, e(\{\mathbf{v}_{k,c}^{(s)}\}_{s=1}^{S})\big)\Big)$
11:             $D_{\text{syn}} \leftarrow D_{\text{syn}} \cup \{(\tilde{x}_i, c)\}$
12:         **end for**
13:     **end for**
14: **end for**
15: Optimize $\phi$ using $D_{\text{syn}}$ by minimizing $\mathbb{E}_{(x,y) \sim D_{\text{syn}}}\big[\ell(f_\phi(x), y)\big]$
1: **LocalTrain**($k$)
2: Initialize LDM with pretrained weights $\theta$ (freeze $\theta$)
3: Obtain stage boundaries $\tau_k = (\tau_{k,0}, \ldots, \tau_{k,S})$ via Local Segmentation($k$)
4: Randomly initialize hypernetwork parameters $\omega_k$
5: **for** $j = 1$ to $Local\ epoch$ **do**
6:     Sample $(x_0, c) \sim D_k, \epsilon \sim \mathcal{N}(0, 1)$, and $t \in \{0, \ldots, T\}$
7:     Compute $z_0 = \varepsilon(x_0)$ and $z_t = \sqrt{\alpha_t}\, z_0 + \sqrt{1 - \alpha_t}\, \epsilon$
8:     Update $\omega_k$ by minimizing $\mathcal{L}_k = \mathbb{E}_{x_0 \sim D_k, \epsilon, t}\Big[\big\|\epsilon_\theta\big(z_t, t \mid e_{\omega,k,c}^{(\tau_k(t))}\big) - \epsilon\big\|_2^2\Big]$
9: **end for**
10: Upload $\tau_k$ and materialized $\{\mathbf{v}_{k,c}^{(s)}\}_{c,\ s=1}^{S}$
1: **Local Segmentation**($k$)
2: Uniformly divide $\{0, \ldots, T\}$ into $m$ blocks, evaluate only at block representatives $\{t_p\}_{p=1}^{m}$
3: **for** $p = 1$ to $m$ **do**
4:     Estimate $\widehat{\ell}_k(t_p) = \frac{1}{M} \sum_{i=1}^{M} \big\|\epsilon_\theta\big(z_{t_p}^{(i)}, t_p \mid e_{\text{base}}\big) - \epsilon^{(i)}\big\|_2^2$
5: **end for**
6: Use $\text{cost}(i, j) = \sum_{p=i}^{j} \big(\widehat{\ell}_k[p] - \mu_{i,j}\big)^2$ to compute segment costs
7: Solve $\min \sum_{s=1}^{S} \text{cost}(b_{s-1} + 1, b_s)$ s.t. $b_s - b_{s-1} \geq \Delta_{\min}$
8: Convert cut points to timestep boundaries $\tau_k = (\tau_{k,0}, \tau_{k,1}, \ldots, \tau_{k,S})$ with $\tau_{k,0} = 0$ and $\tau_{k,S} = T$
9: Define stage mapping $\tau_k(t) = s$ iff $\tau_{k,s-1} \leq t < \tau_{k,s}$
10: **return** $\tau_k$

---

**Computation costs.** The primary computation overhead of FedPrecise comes from optimizing the MLP hypernetwork. However, since the hypernetwork is lightweight, the additional communication overhead introduced by using an MLP (rather than directly optimizing and transmitting descriptors) is marginal. Moreover, in the Local Adaptive Segmentation stage, the MSE estimation over diffusion timesteps does not require backpropagation and only needs a mini-batch sampled from the entire client data. So the extra cost remains acceptable in practice when a client contains multiple categories.

**Synthetic Image Visualization.** We provide representative samples of images generated by FedPrecise across all experimental datasets, which are shown in Figure 8, Figure 9, Figure 10, Figure 11, and Figure 12. Overall, the synthetic images generally recover the characteristic patterns of the corresponding real datasets. In particular, we observe that FedPrecise preserves multiple levels of visual attributes, including layout, content, and material/style cues, while achieving promising perceptual quality.

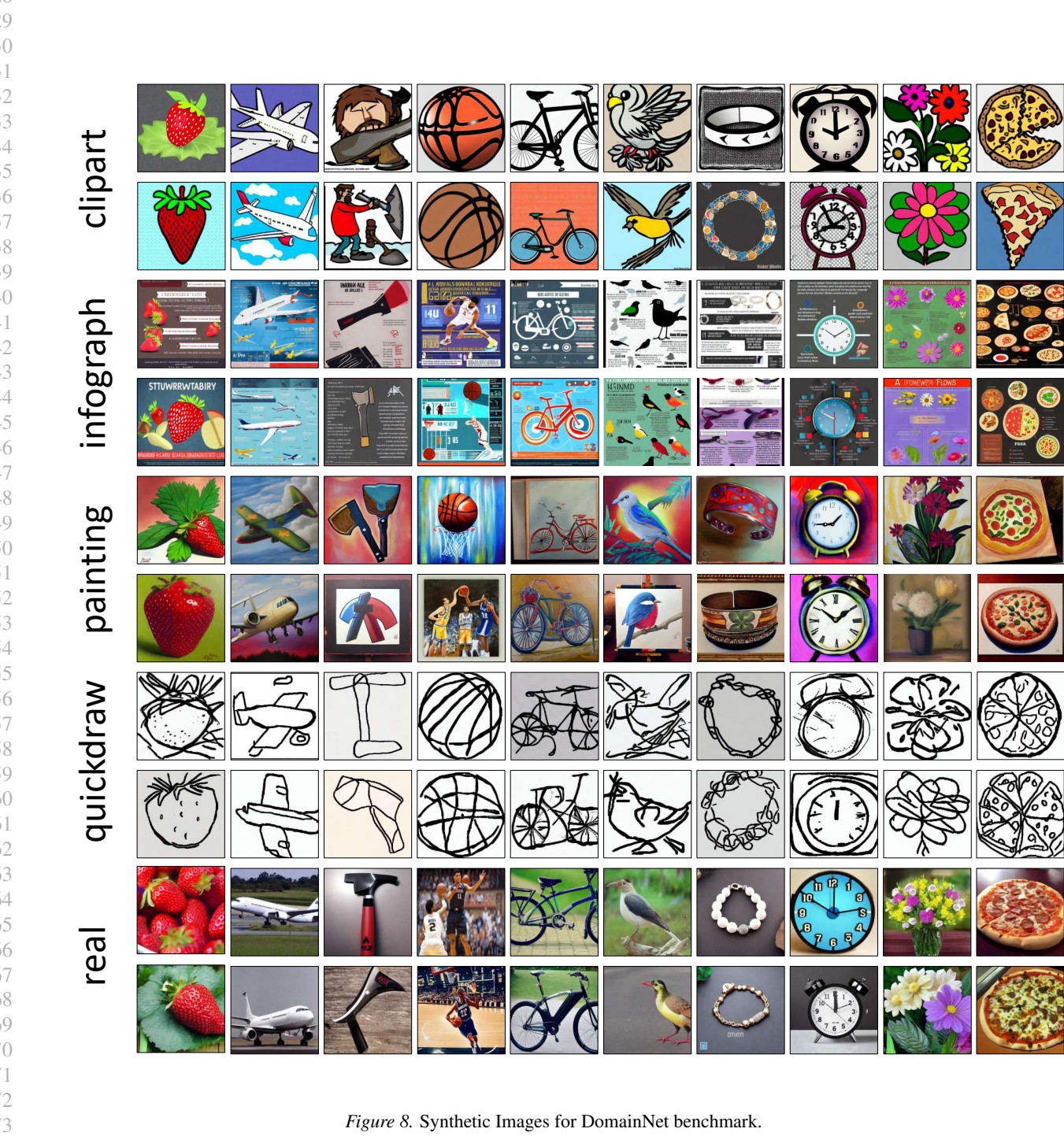

*Figure 8.* Synthetic Images for DomainNet benchmark.

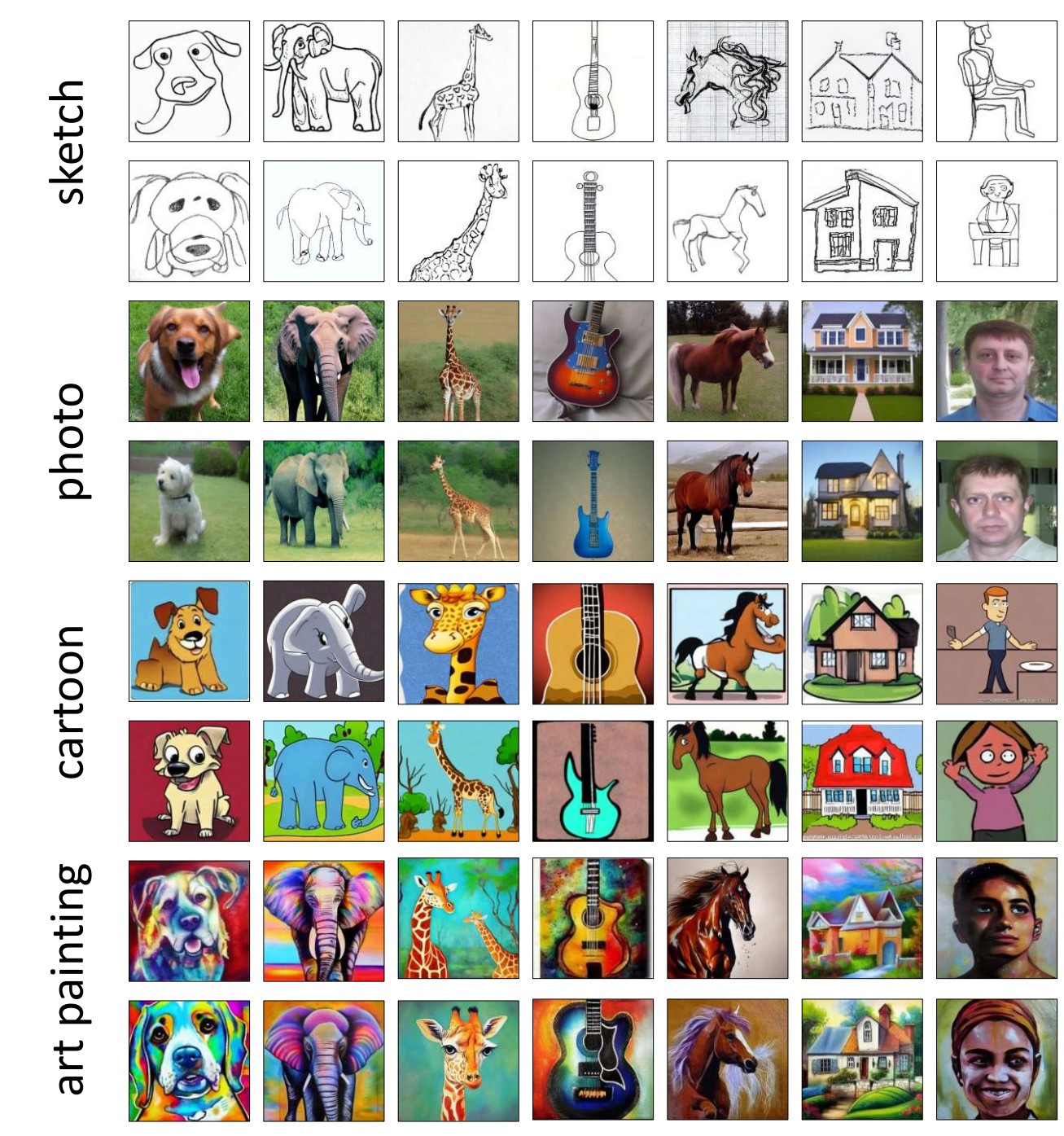

*Figure 9.* Synthetic Images for PACS benchmark.

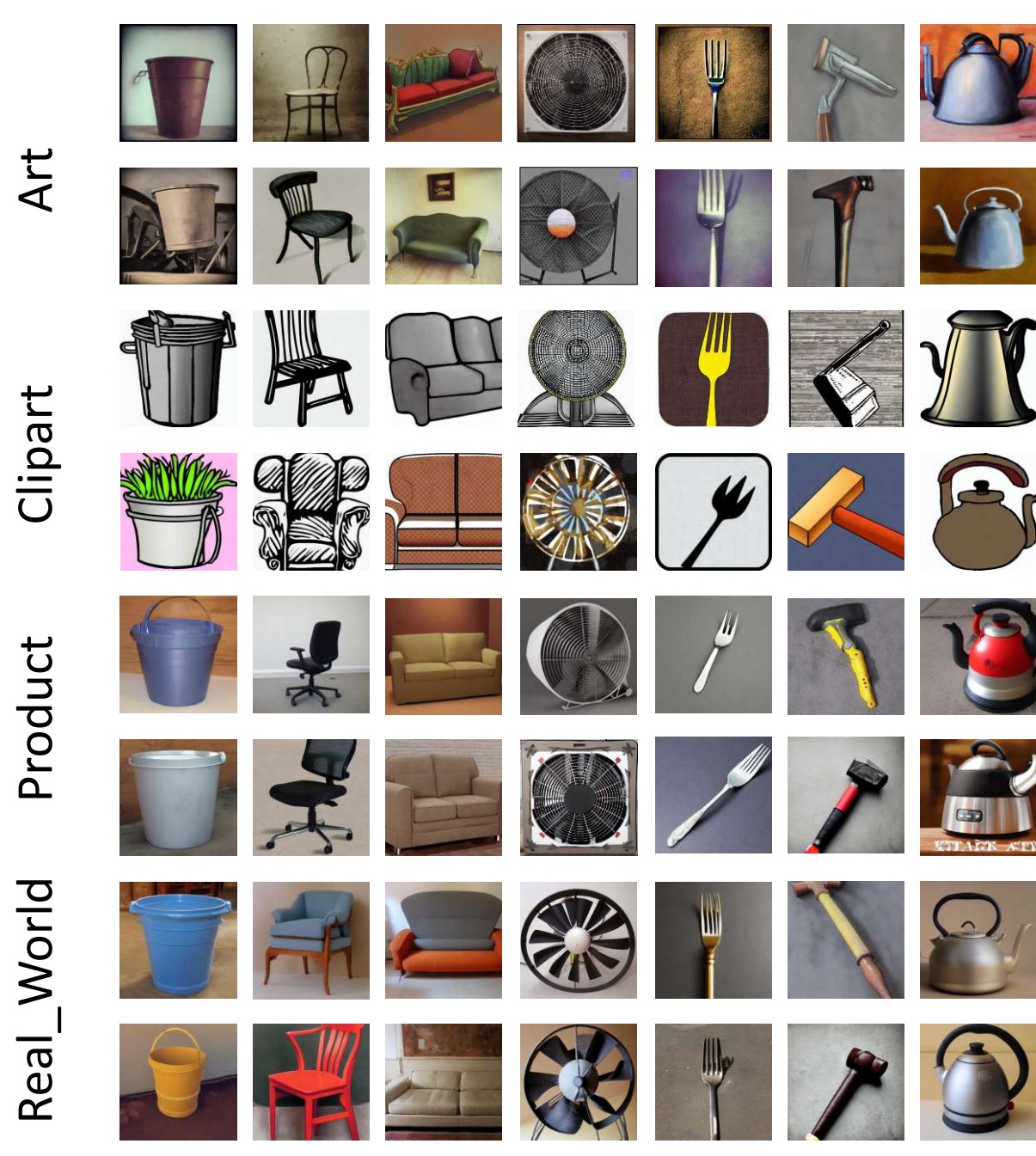

*Figure 10.* Synthetic Images for OfficeHome benchmark.

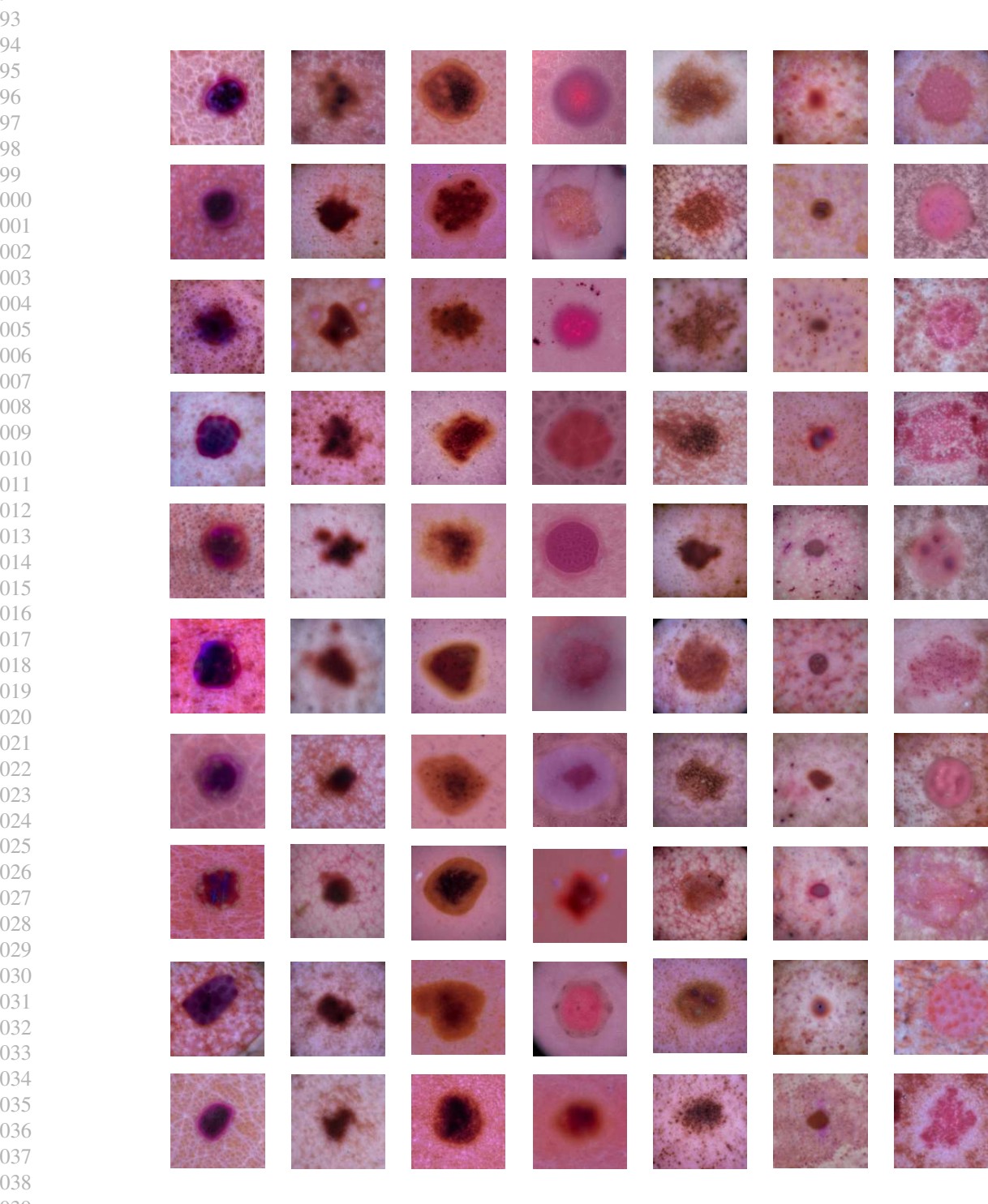

*Figure 11.* Synthetic Images for DermaMNIST benchmark.

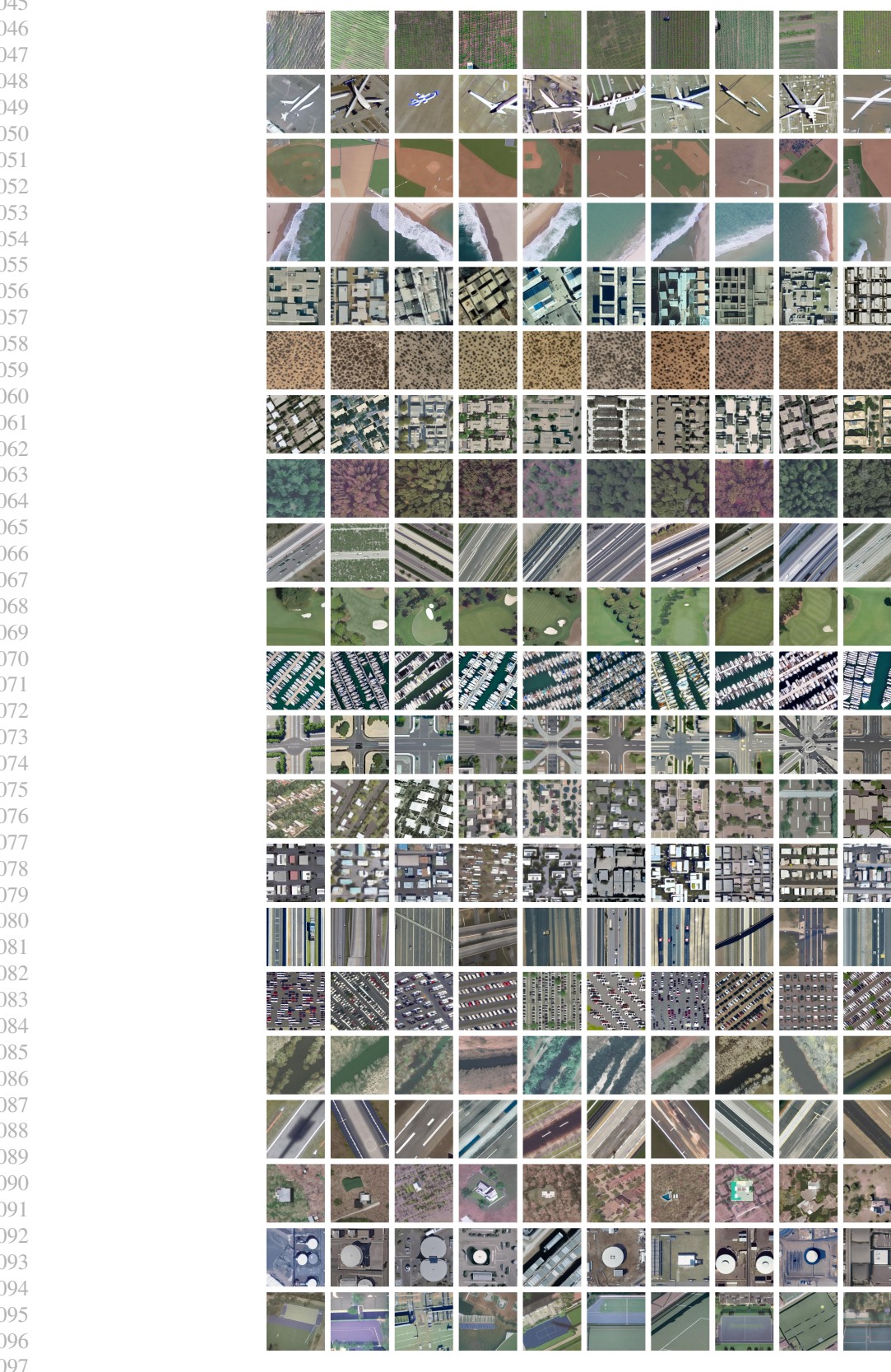

*Figure 12.* Synthetic Images for UCMerced benchmark.

