# OpenReview forum: "FedPrecise: Stage-Precise Diffusion Descriptors for One-Shot Federated Learning"
_ICML.cc/2026/Conference — Submitted to ICML 2026_

### Official Review · Reviewer_8wzm · 2026-03-02

**Soundness:** 3
**Presentation:** 3
**Significance:** 2
**Originality:** 2
**Overall Recommendation:** 3
**Confidence:** 4

**Summary:**

This paper proposes FedPrecise, a one-shot federated learning framework that leverages the stage-wise generation structure of diffusion models to address feature space heterogeneity across clients. The method introduces stage-precise descriptors learned locally and an adaptive timestep segmentation mechanism to guide a latent diffusion model on the server side for high-quality synthetic data generation. The framework achieves strong downstream classification performance across multiple benchmarks with extremely low communication cost. Theoretical analysis provides an upper bound on KL divergence, showing the advantage of multi-stage conditioning over single-stage methods.

**Compliance With Llm Reviewing Policy:**

Affirmed.

**Final Justification:**

The paper proposes an interesting approach to handling feature-space heterogeneity in federated settings and shows promising empirical results. I appreciate that the rebuttal clarifies the novelty relative to prior work, provides more detailed analysis of training cost and memory, and includes additional experimental evidence such as few-shot results and ablations. However, my main concerns regarding scalability to larger and more diverse settings, the reliance on certain theoretical assumptions, and the overall generality of the approach are only partially addressed. In particular, it remains unclear how the method would perform under more challenging real-world scenarios or whether the assumptions consistently hold. As a result, while the rebuttal improves clarity, it does not fundamentally change my assessment, and I recommend a weak reject.

**Key Questions For Authors:**

Could the authors  provide more details on the per-client training time and memory consumption, especially for datasets with many classes ? How does this scale with the number of stages S?

How sensitive is the adaptive segmentation to the number of blocks m and the minimum interval Δmin? Have the authors tested its performance in extreme few-shot settings?

The stage descriptors are learned per client per class. Have the authors considered whether these descriptors could be shared or aggregated across clients to further reduce communication or improve generalization?

Assumption A.2 assumes nonnegative expected log-likelihood gain. Could the authors discuss scenarios where this might not hold and how it would affect the theoretical bound?

**Limitations:**

The authors have provided a brief limitation statement in the Impact Statement section, but it is insufficient. A more detailed discussion of limitations would strengthen the paper.

**Strengths And Weaknesses:**

Strengths:

The paper introduces a conceptually novel idea—leveraging the stage-wise generation process of diffusion models for OSFL. The observation that different diffusion stages correspond to different frequency bands is well-motivated and grounded in prior work.

The proposed method is technically well-designed. The adaptive segmentation  and the MLP-based hypernetwork for descriptor generation are both elegant and effective.

FedPrecise consistently outperforms strong baselines on multiple benchmarks, including real-world datasets like medical and satellite imagery. The improvement is particularly significant in domains with rich multi-stage attributes.

The method achieves state-of-the-art performance while transmitting only 0.768M parameters per client, which is orders of magnitude lower than model-upload methods. This makes it highly practical for real-world deployment.

The paper includes membership inference attack analysis and qualitative visualization, showing that FedPrecise preserves privacy better than baselines.

The paper thoroughly examines the impact of each component, which strengthens the credibility of the claims.

Weaknesses:

While the combination is novel, some components  are adapted from existing works. The paper should more clearly delineate which parts are novel contributions and which are adaptations.

The method requires each client to train an MLP hypernetwork and perform adaptive segmentation for each class. In datasets with many classes, this could become computationally expensive. The paper does not report per-client training time or memory usage.

The adaptive segmentation is based on MSE loss curves estimated from a few blocks. It is unclear whether this approach remains effective when the client data is extremely scarce or when the diffusion model is not well-aligned with the client domain.

Assumption A.2 may not hold in practice if the stage descriptors are poorly optimized or if the client distribution is highly out-of-distribution. The paper could discuss this limitation more explicitly.

Limited Discussion on Negative Societal Impact: The paper only includes a one-sentence statement ("none of which we feel must be specifically highlighted here"). Given the potential misuse of generative models (e.g., deepfakes) and privacy risks in federated learning, a more detailed discussion would be beneficial.

---

> ### Author Rebuttal · Authors · 2026-03-31
>
> We sincerely appreciate the reviewer’s thoughtful review and helpful comments!
>
> > **Response to weakness about clarity of novel contributions**
>
> We acknowledge that some components of FedPrecise are inspired by prior works. In particular, our work builds upon foundational diffusion-based OSFL methods such as FedDEO, while ProSpect provides evidence that different diffusion stages correspond to different visual attributes.
>
> However, we identify that prior works such as FedDEO and FedBiP do not sufficiently address feature-space heterogeneity. FedPrecise mitigates this issue by using stage-wise conditioning to more accurately reconstruct client-specific characteristics in synthesized data. To fit the lightweight client setting, we replace ProSpect’s heavy multi-layer attention hypernetwork with a newly designed MLP, making the joint training of multi-stage descriptors feasible. In addition, we propose a lightweight adaptive segmentation method to further improve the ability of FedPrecise to capture heterogeneous client features, which is also not explored in prior work.
>
>
>
> > **Response to question about training time and memory**
>
> We regret any confusion caused by our presentation. In FedPrecise, adaptive segmentation captures shared characteristics within a client, and therefore only needs to be performed once per client rather than per class. This design significantly reduces the additional computational overhead.
>
> Using a single NVIDIA RTX 3090 GPU (24GB memory) to simulate the full FedPrecise pipeline (24-shots, 6 clients on DomainNet), the peak GPU memory usage is 3.85 GB, peak CPU memory usage is 3.04 GB, and the wall-clock time is 12 h 36 min.
>
> To further quantify the cost, we report per-client, per-class FLOPs for FedLMG, FedDEO, FedBiP, FedPrecise, and CLIP-based federated fine-tuning (for reference):
>
> |method|CLIP-based FT|FedLMG|FedDEO|FedBiP|FedPrecise|
> |---|---:|---:|---:|---:|---:|
> |computation cost（GFLOPs）|692.58|196.01|513.25|561.33|540.68|
>
> As shown by the results, FedPrecise introduces only about 5.07% additional computation compared to FedDEO, lower than the extra cost of FedBiP due to additional text encoder passes and is acceptable. However, although the average computational cost per class does not change significantly with the increase of stage number $S$, the total computation on the client will inevitably grow as the number of classes increases.
>
>
>
> > **Response to question about segmentation sensitivity and few-shot performance**
>
> In Section 4.2, DermaMNIST and UCMerced are regarded as rare domains scarcely covered by the pretrained LDM. FedPrecise outperforms all baselines on these datasets. As shown in Figure 4, it visibly reconstructs client-specific styles more faithfully than the current SOTA method FedBiP.
>
> To further demonstrate the effectiveness of adaptive segmentation, we provide results under 4-shot and 1-shot settings, as well as ablation studies of segmentation component. Sensitivity analysis is discussed in our response to **Reviewer b4ib**. Results on PathMNIST and OpenEarthMapSAR, where the diffusion model is worse aligned, are provided in our response to **Reviewer QpDV**.
>
> |shots|FedPrecise|FedPrecise w/o Seg|FedBiP|FedLMG|FedDEO|
> |---|---:|---:|---:|---:|---:|
> |4-shot|69.28|67.05|65.35|64.08|62.37|
> |1-shot|66.95|63.11|63.18|58.67|60.92|
>
>
>
> > **Response to question about aggregating stage descriptors**
>
> We have also considered this question during the development of FedPrecise. However, in our method, descriptors are not treated as aggregatable model parameters; instead, they serve as intermediate guidance for data generation.
>
> Therefore, avoiding aggregation helps preserve client-specific characteristics in the generated data, while aggregating descriptors may blur heterogeneous features and distort the data distribution. Nevertheless, we believe this is still a worthwhile direction for future exploration.
>
>
>
> > **Response to question about Assumption**
>
> Assumption A.2 in the theoretical analysis of FedPrecise is based on a reasonable scenario: in a K-stage denoising process, compared to the case without conditioning at stage k, introducing the k-th descriptor leads to a nonnegative expected log-likelihood gain. Since each descriptor is trained specifically for its corresponding denoising stage, this assumption is generally well justified.
>
> However, in certain extreme cases, such as when descriptor training does not converge, this assumption may not hold. In such cases, the upper bound
> $$
> \mathrm{KL}(p_n(x)\,\|\,p_{\epsilon_\theta}(x \mid d^{1:K})) \le \lambda - \sum_{k=1}^{K} \gamma_k
> $$
>
> becomes bigger depending on the value of
> $$
> \sum_{k=1}^{K} \gamma_k
> $$
> In other words, the more descriptors misguide the generation compared to the unconditioned case, the looser the upper bound becomes.
>
>
>
> > **Response to limitation**
>
> We will expand the impact statement to include discussions on the generative models and privacy risks in the revised version.

---

> > ### Author Rebuttal · Reviewer_8wzm · 2026-04-03
> >
> > Thank you for the detailed rebuttal. I have a few remaining questions: it would be helpful to better understand how the method generalizes under more extreme distribution shifts or larger-scale settings (more clients/classes), as well as its practical scalability given that total computation grows with class count. I am also interested in whether hybrid strategies for descriptor sharing could balance personalization and generalization, and in how sensitive the method is when the key theoretical assumption does not hold in practice. These clarifications would further strengthen the work.

---

> > > ### Author Response · Authors · 2026-04-06
> > >
> > > We are sincerely grateful for your careful reading of our rebuttal and for your thoughtful comments, which have been very helpful in clarifying our work.
> > >
> > > > **Response to question about generalization under more extreme distribution shifts**
> > >
> > > In **Section 4.3**, we report FedPrecise on two rare datasets with severe distribution shifts, DermaMNIST and UCMerced. Results on the more challenging PathMNIST and OpenEarthMapSAR are provided in our response to **reviewer QpDV**. The results show that FedPrecise outperforms all baselines on highly diverse domains, including skin cancer, color satellite, H&E-stained pathology, and radar images, all substantially different from the LDM pretraining data, and that this advantage remains consistent across label heterogeneity levels.
> > >
> > > To further demonstrate FedPrecise under more extreme distribution shifts, we additionally provide results on binary classification over the multi-domain breast cancer dataset Camelyon17, where different levels of label heterogeneity are simulated as well by controlling per-domain sample sizes:
> > >
> > > |Alpha|FedAvg|FedPrecise|FedBiP|FedDEO|FedLMG|
> > > |---|---|---|---|---|---|
> > > |5.0|85.65|86.68|81.39|78.59|70.10|
> > > |0.5|81.97|80.74|75.32|74.02|66.65|
> > > |0.01|79.46|77.38|73.09|72.94|64.73|
> > >
> > > |Method \ Score(×10^-2)|MMD↓|CMMD↓|
> > > |---|---|---|
> > > |FedDEO|23.81|15.53|
> > > |FedPrecise|18.26|12.18|
> > >
> > > The MMD/CMMD results indicate that on datasets with extreme distribution shifts, FedPrecise generates images closer to the real distribution than the representative diffusion-based baseline FedDEO. This improved fidelity leads to better downstream classification and further demonstrates the generalization of our method.
> > >
> > > > **Response to question about scalability of clients**
> > >
> > > To evaluate the generalization ability of FedPrecise in larger-scale OSFL scenarios, we provide scalability experiments on DomainNet.
> > >
> > > |Method \ Clients|6|12|18|24|30|
> > > |---|---:|---:|---:|---:|---:|
> > > |**FedPrecise**|76.21|79.43|81.71|83.54|85.71|
> > > |**FedAvg**|71.26|74.09|76.07|77.88|79.56|
> > > |**Central**|81.89|85.36|87.62|88.83|89.47|
> > >
> > > The results show that FedPrecise consistently improves as the number of clients increases, similar to prior diffusion-based methods, further demonstrating its generalization ability.
> > >
> > > > **Response to question about scalability of classes and computation cost**
> > >
> > > First, to evaluate FedPrecise on a larger number of classes, we provide scalability experiments with respect to class count.
> > >
> > > |Method \ Classes|30|20|10|
> > > |---|---|---|---|
> > > |Central|64.39|70.86|81.89|
> > > |FedPrecise|61.53|64.20|76.21|
> > > |FedAvg|47.68|57.87|71.26|
> > >
> > > The results show that, as the number of classes increases, the classification accuracy of FedPrecise naturally declines. At the same time, its advantage over FedAvg becomes increasingly pronounced, which demonstrates the promising scalability of the method. Moreover, although the computational cost of FedPrecise, like that of FedDEO and FedBiP, inevitably increases with the number of classes, FedPrecise consistently maintains a lower per-class computational cost than the current state-of-the-art method FedBiP. In addition, its peak GPU memory usage of only 3.85 GB also makes FedPrecise easier to deploy on client devices.
> > >
> > > > **Response to question about hybrid strategies for descriptor sharing**
> > >
> > > To evaluate whether descriptor sharing improves generalization, we aggregate same-stage descriptors by domain/class, and then use the aggregated descriptors to replace the text embedding and guide image synthesis together with the original descriptors. The results are as follows:
> > >
> > > |Agg_method|Q|S|C|R|I|P|Acc|
> > > |---|---|---|---|---|---|---|---|
> > > |domain|62.71|77.66|82.79|86.87|58.44|70.81|73.21|
> > > |class|63.87|81.24|84.01|86.15|59.13|70.15|74.09|
> > >
> > > The experimental results show that, at both the within-domain and cross-domain class levels, the descriptor aggregation strategy fails to bring better generalization to FedPrecise.
> > >
> > > > **Response to question about practical assumption violation**
> > >
> > > To simulate scenarios in which Assumption A.2 does not hold, we randomly replace several stages in the fixed ten-stage descriptors of FedPrecise with descriptors from different classes, so that the conditions at certain stages become misleading, thereby violating the assumption:
> > >
> > > |Mis_stage|Q|S|C|R|I|P|Acc|
> > > |---:|---:|---:|---:|---:|---:|---:|---:|
> > > |1|66.67|80.41|85.42|87.92|58.51|72.92|75.31|
> > > |2|62.47|80.48|88.23|89.06|58.22|70.16|74.77|
> > > |3|64.60|76.83|84.61|88.05|53.94|73.58|73.60|
> > > |4|63.60|80.14|85.42|87.20|53.94|69.88|73.36|
> > > |5|55.93|77.93|83.81|87.53|53.94|71.20|71.72|
> > >
> > > The experimental results show that, as the number of misleading stages increases, the performance of FedPrecise degrades gradually and mildly. In particular, when fewer than two stages are corrupted, the degradation remains quite limited. This suggests that even when Assumption A.2 is violated, the method still maintains stable performance and exhibits a certain degree of robustness to mild violations of this assumption.

---

### Official Review · Reviewer_QpDV · 2026-03-11

**Soundness:** 3
**Presentation:** 3
**Significance:** 2
**Originality:** 2
**Overall Recommendation:** 4
**Confidence:** 3

**Summary:**

In one-shot federated learning (OSFL), the paper proposes FedPrecise, a method in which each client, instead of transmitting the entire prompt or model, learns separate descriptors for different denoising stages of the diffusion process and uploads them to the server, which then uses them to generate synthetic data.

**Compliance With Llm Reviewing Policy:**

Affirmed.

**Final Justification:**

The authors have addressed some of my concerns through additional experiments. However, technically, it still remains unclear why FedPrecise inherently outperforms potential backbone fine-tuning approaches (e.g., LoRA) or other methods. The analysis is insufficient, and I maintain my original score.

**Key Questions For Authors:**

* What is the per-client compute and memory footprint for (a) adaptive segmentation and (b) descriptor training? Please report FLOPs, wall-clock times, and peak memory on a representative GPU/CPU.

* Why are the Central/FedAvg performances higher than those reported in other papers?

* FedBiP is sensitive to the number of synthetic samples. How many synthetic samples were used in this paper?

* How does FedPrecise perform when the diffusion backbone is severely out-of-domain (e.g., histopathology or SAR imagery)? Would limited fine-tuning on server with DP be feasible?

* What is the exact base embedding $e_{base}$ used to compute denoising-difficulty curves, and how sensitive is segmentation to this choice?

* Can you report statistical significance (mean $\pm$ std over multiple runs) for the main tables?

**Strengths And Weaknesses:**

**Strengths**
* The paper is well written, and the problem statement is clear and well motivated.
* Figure 1 effectively illustrates that FedBiP applies the same conditioning across all diffusion timesteps, which may limit its ability to capture stage-specific variations in the denoising process. This figure helps clearly motivate the proposed approach.
* Benchmarks are appropriate and multiple domains are covered. Improvements over strong OSFL baselines (FedDEO, FedBiP, FedLMG) are consistent. Visuals in medical/satellite domains are compelling and align with the qualitative motivation. FedPrecision consistently outperforms existing approaches, and the broad experimental evaluation provides strong empirical evidence for the effectiveness of the proposed method.

**Weakness**

* Client-side computation may be non-trivial, as training stage-wise descriptors appears to require backpropagation through the LDM and text encoder. However, the paper does not quantify the resulting FLOPs, memory usage, or runtime on the client side.

* The paper provides limited statistical reporting. Confidence intervals or variance across multiple runs are not reported, making it difficult to assess the robustness of the results, particularly given that some improvements are relatively small (≈2–3%).

* The paper evaluates privacy risks using membership inference attacks. However, diffusion-based generative models may be vulnerable to other types of attacks (e.g., reconstruction attacks or attribute inference). Evaluating additional attack scenarios would provide a more comprehensive understanding of the privacy properties of the proposed approach.

**Minor weakness**

* Some equations and symbols contain typos that impede exact understanding (e.g., Eq. 15 sampling, minor indexing and brace errors). It would be helpful if the statements in the proof were written more clearly and accessibly.
* The appendix reference at the very end of the “Baseline methods settings” section appears to be broken.

---

> ### Author Rebuttal · Authors · 2026-03-31
>
> We appreciate the reviewer’s detailed comments and valuable suggestions!
>
> > **Response to weakness about privacy**
>
> We further report additional privacy evaluations under reconstruction attacks [1] and attribute inference [2] for FedBiP and FedPrecise, where FedAvg is included for reference:
>
> |method|property_inference|reconstruction|
> |---|---:|---:|
> |FedPrecise|0.18|0.37|
> |FedBiP|0.52|0.43|
> |FedAvg|0.25|0.22|
>
> The results show that, since FedPrecise generates images that are close to local data at the feature level rather than the instance level, it achieves better privacy performance than FedBiP, which uploads latent information of original images.
>
>
>
> > **Response to question about computation cost**
>
> For the computation details of FedPrecise, please refer to our response to **Reviewer 8wzm**. Specifically, (a) the computation cost of adaptive segmentation is 27.34 GFLOPs, and (b) the computation cost of descriptor training is 513.34 GFLOPs. This is because segmentation requires additional forward passes through the UNet, whose cost is much higher than training the extra MLP.
>
> However, owing to the lightweight design of the MLP and the fact that each client performs adaptive segmentation only once, FedPrecise requires only about 5.07% additional computation compared with FedDEO, which is still lower than FedBiP, where an extra pass through the text encoder is required in every training iteration, making computation overhead in FedPrecise acceptable. Meanwhile, the peak GPU memory usage of no more than 4 GB makes deployment on the client side feasible.
>
>
>
> > **Response to question about higher Central/FedAvg performance**
>
> In our experimental setting, the global classifier is consistently a ResNet-18 pretrained on ImageNet. In addition, DomainNet and PACS are evaluated under the 24-shots setting, which differs from the unpretrained ResNet-18 results reported in FedDEO and also from the 16-shots setting used in FedBiP. A stronger classifier backbone and a larger number of local images can both improve the performance of Central and FedAvg.
>
> In addition, we further evaluated FedPrecise under the extreme few-shot setting; please refer to our response to **Reviewer 8wzm**. The results show that FedPrecise still maintains its advantage in this regime, and that adaptive segmentation remains effective.
>
>
>
> > **Response to question about number of synthetic samples**
>
> As stated in the Baseline Methods of Appendix, for fairness and efficiency, both FedPrecise and FedBiP use the setting of generating 80 images per domain per class in this paper.
>
>
>
> > **Response to question about out-of-domain and DP fine-tuning**
>
> We further evaluate FedPrecise on two datasets where the diffusion backbone is severely out-of-domain, namely PathMNIST and OpenEarthMapSAR:
>
> |Dataset|alpha|Fedavg|FedDEO|FedBiP|FedLMG|FedPrecise|
> |---|---|---:|---:|---:|---:|---:|
> |PathMNIST|5.00|63.08|60.03|64.75|43.16|66.32|
> |PathMNIST|0.50|57.45|55.53|56.86|37.52|59.05|
> |PathMNIST|0.01|54.38|53.36|55.69|35.68|58.99|
> |OpenEarthMapSAR|5.00|51.31|45.12|49.63|36.92|51.38|
> |OpenEarthMapSAR|0.50|43.94|41.17|44.03|33.46|48.25|
> |OpenEarthMapSAR|0.01|39.83|33.98|39.40|32.33|42.08|
>
> Following the setup in Section 4.2, we partition each dataset into five clients with different levels of label heterogeneity controlled by α. The results indicate that FedPrecise can better guide the server to generate such rare-domain data even under backbone–data distribution mismatch.
>
> In principle, adapting the backbone via fine-tuning methods such as LoRA could help the server better handle domain mismatch. However, this would introduce additional computational overhead and potentially higher privacy risks, even with differential privacy. Therefore it is beyond the scope of our work. Still, we believe this is a worthwhile direction for future research.
>
>
>
> > **Response to question about base embedding**
>
> Please refer to our response to **Reviewer b4ib**, where sensitivity analyses on the base embedding demonstrate the stability of the segmentation.
>
>
>
> > **Response to question about statistical significance**
>
> In this paper, all results reported in Table 1, Table 2, and throughout the rebuttal are averaged over three random seeds. However, due to space constraints, we regret that we are only able to provide the main-table results with standard deviations for three representative methods of DomainNet in this response. We will provide the complete statistics in the revised version.
>
> |Method|C|I|P|Q|R|S|Accuracy|
> |---|---:|---:|---:|---:|---:|---:|---:|
> |**FedPrecise**|87.36±0.58|58.08±1.12|72.80±0.66|69.42±1.34|87.80±0.39|81.81±0.61|76.21±0.47|
> |**FedDEO**|78.93±0.74|51.74±1.08|71.01±0.69|70.79±1.27|84.40±0.52|74.56±0.83|71.90±0.56|
> |**FedBiP**|82.11±0.67|57.96±1.24|71.25±0.73|67.89±1.41|86.33±0.46|70.22±0.78|72.63±0.53|
>
> [1] Deep Leakage from Gradients, NeurIPS 2019.
> [2] Model Inversion Attacks that Exploit Confidence Information and Basic Countermeasures, ACM CCS 2015.

---

> > ### Author Rebuttal · Reviewer_QpDV · 2026-04-03
> >
> > I would like to thank the authors for their detailed response and the inclusion of additional experimental results. Some of the weaknesses and questions have been addressed. However, there are still unresolved problems.
> >
> > While the experimental results on PathMNIST and OpenEarthMapSAR are promising, the technical justification for why FedPrecise inherently outperforms potential backbone fine-tuning (e.g., LoRA) or other methods remains unclear.
> >
> > The reconstruction and membership inference attack methods used in the evaluation appear to be outdated. It would strengthen the paper to reevaluate the method using more recent attack techniques. For gradient inversion attacks (GIA), please consider [1,2], and for membership inference attacks (MIA), please consider [3,4]. If the source codes for these methods are not publicly available, it would still be desirable to evaluate the method using other recent state-of-the-art attacks.
> >
> > [1] Fang, Hao, et al. “GIFD: A Generative Gradient Inversion Method with Feature Domain Optimization.” Proceedings of the IEEE/CVF International Conference on Computer Vision, 2023.
> >
> > [2] Meng, Jiayang, et al. “Is Diffusion Model Safe? Severe Data Leakage via Gradient-Guided Diffusion Model.” arXiv preprint arXiv:2406.09484, 2024.
> >
> > [3] Li, Jiacheng, Ninghui Li, and Bruno Ribeiro. “Effective Passive Membership Inference Attacks in Federated Learning against Overparameterized Models.” The Eleventh International Conference on Learning Representations, 2023.
> >
> > [4] Zhu, Gongxi, et al. “FedMIA: An Effective Membership Inference Attack Exploiting ‘All for One’ Principle in Federated Learning.” Proceedings of the IEEE/CVF Conference on Computer Vision and Pattern Recognition, 2025.

---

> > > ### Author Response · Authors · 2026-04-06
> > >
> > > We sincerely thank you for your valuable suggestions and comments on our response!
> > >
> > > > **Response to question about comparison with backbone fine-tuning methods**
> > >
> > > To further evaluate the effect of backbone fine-tuning methods, we followed [1] on the DomainNet dataset and applied the representative LoRA method C-LoRA to fine-tune the server-side diffusion model on the K and V projections of the UNet cross-attention layers. A separate low-rank adapter was trained for each class in each domain, with the rank set to 16. We also followed [2] to evaluate the DP-LoRA method with differential privacy:
> > >
> > > |Dataset|Q|S|C|R|I|P|Acc|
> > > |---|---|---|---|---|---|---|---|
> > > |FedPrecise|69.42|81.81|87.36|87.80|58.08|72.80|76.21|
> > > |C-LoRA|63.42|78.77|79.84|88.94|60.15|76.65|74.63|
> > > |DP-LoRA|61.88|78.34|78.21|87.56|57.47|75.02|73.08|
> > >
> > > Regarding privacy, please refer to our evaluation of the above two LoRA-based methods in the **Response to question about evaluating privacy with more recent attack methods**. These results show that although C-LoRA can outperform some diffusion-based baselines, FedPrecise still achieves better overall performance. Moreover, under both MIA and GIA attacks, DP-LoRA does not demonstrate a significant advantage over FedPrecise. This further suggests that multi-stage descriptors are a more efficient choice for personalization than backbone fine-tuning, especially considering that the communication overhead of C-LoRA reaches 159.130M parameters, whereas FedPrecise requires only 0.768M in the same setting, which greatly reduces the communication burden when scaling to large-scale scenarios.
> > >
> > >
> > >
> > >
> > > > **Response to question about evaluating privacy with more recent attack methods**
> > >
> > > We followed [3] and [4] to re-evaluate FedPrecise, FedBiP, and the two LoRA-based methods under both MIA and GIA. Since diffusion-based methods only upload descriptors rather than model parameters under a one-shot budget, for MIA we used the descriptors/low-rank adapters as the measurement target for a single-round attack, and adopted AUC as the evaluation metric. For the same reason, in GIA, we did not adopt the gradient optimization process used in the original paper, and instead used the exposed descriptors/low-rank adapters to perform excessive sampling with the original diffusion model, measuring the quality of reconstructed attack data with SSIM.
> > >
> > > | Method     | MIA/AUC↓ | GIA/SSIM↓ |
> > > | ---------- | -------- | --------- |
> > > | FedPrecise | 0.53     | 0.57      |
> > > | FedBiP     | 0.80     | 0.64      |
> > > | C-LoRA     | 0.63     | 0.69      |
> > > | DP-LoRA    | 0.55     | 0.55      |
> > > | FedAvg     | 0.66     | 0.95      |
> > >
> > > The experimental results further show that FedPrecise demonstrates stronger resistance than FedBiP under both attack settings, and remains highly competitive even compared with DP-LoRA, which is equipped with differential privacy.
> > >
> > >
> > >
> > > We once again thank you for thoughtful feedback and guidance on the privacy evaluation, which has provided us with a valuable opportunity to further clarify our work. We hope our response has addressed your concerns.
> > >
> > > [1] Smith, James Seale, et al. “Continual Diffusion: Continual Customization of Text-to-Image Diffusion with C-LoRA.” arXiv preprint arXiv:2304.06027, 2023.
> > >
> > > [2] Tsai, Yu-Lin, et al. “Differentially Private Fine-Tuning of Diffusion Models.” Proceedings of the IEEE/CVF International Conference on Computer Vision, 2025, pp. 4561–4571.
> > >
> > > [3] Zhu, Gongxi, et al. “FedMIA: An Effective Membership Inference Attack Exploiting ‘All for One’ Principle in Federated Learning.” Proceedings of the IEEE/CVF Conference on Computer Vision and Pattern Recognition, 2025.
> > >
> > > [4] Meng, Jiayang, et al. “Is Diffusion Model Safe? Severe Data Leakage via Gradient-Guided Diffusion Model.” arXiv preprint arXiv:2406.09484, 2024.

---

### Official Review · Reviewer_b4ib · 2026-03-12

**Soundness:** 3
**Presentation:** 3
**Significance:** 4
**Originality:** 3
**Overall Recommendation:** 5
**Confidence:** 4

**Summary:**

This paper introduces FedPrecise, a novel method for addressing the challenges of feature space heterogeneity in One-Shot Federated Learning (OSFL). The authors point out the limitations of single conditioning in existing diffusion-based OSFL methods, which overlooks the stage-wise nature of image synthesis and results in significant distribution shifts in synthetic data. Therefore,  FedPrecise focuses on a central concept of stage-precise personalization via compact token descriptors and a local adaptive segmentation mechanism. By learning and transmiting lightweight, stage-specific tokens, FedPrecise enables the server-side diffusion model to more accurately reconstruct diverse client distributions. Experiments on diverse benchmarks demonstrate that FedPrecise consistently outperforms state-of-the-art baselines in classification accuracy while maintaining significantly lower communication overhead.

**Compliance With Llm Reviewing Policy:**

Affirmed.

**Final Justification:**

The author provided extensive experimental results in response to my questions. I acknowledge the contributions made by this paper. Therefore, I am willing to raise my score by one point to support the author.

**Key Questions For Authors:**

1.In Sec. 3.2 (Eq. 6 and 7), the method calculates the local denoising-difficulty curve using a fixed base embedding $e_{\text{base}}$. Is it an unconditional null embedding or a generic text prompt like "an image"?

2.The experiments use 8-shot or 24-shot settings. Can the hypernetwork still effectively learn a multi-stage descriptor in scenarios that clients have less samples (e.g., 4-shot)?

3.Does the MLP hypernetwork require any specific optimizer, weight regularization or dropout during the training of descriptors? As the optimization relevant to diffusion models often requires more complex optimizer (e.g., AdamW), such details are important.

**Limitations:**

The authors can discuss the limitations about dependency on pre-trained diffusion models or computation costs.

**Strengths And Weaknesses:**

**Strengths**

1.The paper is well-structured with clear visualizations, which makes the motivation of stage-wise segmentation and the mechanisms of diffusion-based learning clear.

2.The authors present a core question regarding single conditioning in diffusion-based OSFL fails to capture stage-specific semantics. The proposed stage-precise descriptors combined with local adaptive segmentation offers an original and theoretically sound solution for it in OSFL.

3.Extensive experiments demonstrate that FedPrecise consistently outperforms SOTA baselines on challenging heterogeneous scenarios, such as domain-shift, medical and satellite imaging datasets, which demonstrate the adaptability to different types of datasets. Moreover, the communication overhead caused by FedPrecise is also acceptable.

**Weaknesses**

1.The method utilizes fixed prompt templates, e.g., "a [domain] of [category]". The paper does not explore whether the training of descriptors are sensitive to such templates. For instance, it is unclear whether more descriptive template would affect the ability of the descriptors to capture stage-wise features.

2.The Local Adaptive Segmentation mechanism divides the diffusion process into $m=100$ blocks to estimate the difficulty curve. The paper does not discuss the sensitivity of the performance to this hyperparameter. Moreover, it is unclear how to choose the stochastic evaluations $M$.

3.The current evaluation is limited to small number of clients (one per domain). It is unclear how FedPrecise would perform in large-scale OSFL scenarios with more clients.

---

> ### Author Rebuttal · Authors · 2026-03-31
>
> We thank the reviewer for the careful reading and constructive feedback!
>
> > **Response to question about prompt template and base embedding**
>
> In this paper, the base embedding used in FedPrecise is "A photo of an object." We conduct sensitivity analyses on DomainNet using several different base embeddings and different embedding templates:
>
> |base_embedding|Q|S|C|R|I|P|accuracy|
> |---|---:|---:|---:|---:|---:|---:|---:|
> |an image|70.36|82.32|85.06|86.43|59.71|71.78|75.94|
> |a picture of an object|72.06|80.28|85.25|87.38|58.49|72.67|76.02|
> |a natural image of an object|74.12|80.03|84.48|86.84|57.06|72.29|75.80|
> |a real-world image of an object|73.61|82.90|84.48|85.71|58.49|69.86|75.84|
>
>
>
> |templates|Q|S|C|R|I|P|accuracy|
> |---|---:|---:|---:|---:|---:|---:|---:|
> |a [domain] image of [category]|70.36|82.32|85.06|86.43|59.71|71.78|75.94|
> |an image of [category] in [domain] style|71.15|80.41|84.48|87.44|57.87|73.44|75.80|
> |a depiction of [category] in [domain] style|73.03|80.03|84.67|87.26|57.67|72.54|75.87|
> |a real-world image of [category] in [domain] style|72.82|79.75|84.91|86.42|58.46|72.01|75.73|
>
> The results show that FedPrecise is robust to the choice of both common base embedding and embedding template, and maintains strong performance across different settings.
>
>
>
> > **Response to weakness about sensitivity in Local Adaptive Segmentation**
>
> In FedPrecise, the hyperparameters of the adaptive segmentation are set as $m = 100$, $M = 40$, $Delta_{min} = 20$, and $S = 10$, where $M$ is chosen as a trade-off between computational cost and performance. Sensitivity analysis for the stage number S is already provided in Table 4, and we further conduct sensitivity analyses for all other parameters:
>
> |m|M|$Delta_{min}$|S|Q|S|C|R|I|P|accuracy|
> |---|---:|---:|---:|---:|---:|---:|---:|---:|---:|---:|
> |100|40|40|10|58.48|83.30|85.33|88.55|61.10|74.67|75.24|
> |100|40|10|10|59.27|80.15|86.02|86.55|58.90|73.69|74.10|
> |100|20|20|10|62.70|82.55|87.21|88.02|58.26|73.67|75.40|
> |100|10|20|10|55.03|80.28|85.82|86.49|59.92|72.92|73.41|
> |200|40|20|10|60.45|84.68|86.53|89.67|59.85|74.14|75.89|
> |50|40|20|10|66.18|81.17|86.40|86.73|57.46|70.37|74.72|
>
> The results show that FedPrecise does not exhibit severe performance degradation except when $M = 10$ or $Delta_{min} = 10$, corresponding to severely insufficient sampling budget or a minimum segment length smaller than the diffusion sampling step. This demonstrates the robustness of the adaptive segmentation module to these hyperparameters.
>
>
>
> > **Response to weakness about scalability to more clients**
>
> To evaluate the performance of FedPrecise in larger-scale OSFL scenarios, scalability experiments are provided.
>
> |Method \ clients per domain|1|2|3|4|5|
> |---|---:|---:|---:|---:|---:|
> |**FedPrecise**|76.21|79.43|81.71|83.54|85.71|
> |**FedAvg**|71.26|74.09|76.07|77.88|79.56|
> |**Central**|81.89|85.36|87.62|88.83|89.47|
>
> The results show that the performance of FedPrecise consistently improves as the number of clients increases, similar to prior diffusion-based methods, further demonstrating the effectiveness of FedPrecise in larger-scale OSFL scenarios.
>
>
>
> > **Response to question about performance under fewer-shot settings**
>
> To evaluate whether FedPrecise can effectively learn under fewer samples, we assess its performance along with several representative diffusion-based OSFL methods under 4-shot and 1-shot settings in our response to **Reviewer 8wzm**. The results show that FedPrecise can still efficiently learn descriptors even with extremely limited client data.
>
>
>
> > **Response to question about MLP training details**
>
> In FedPrecise, our MLP hypernetwork is optimized using the AdamW optimizer, with L2 regularization applied to stabilize training. We also employ a dropout rate of 0.1 to prevent overfitting during the descriptor learning process.
>
>
>
> > **Response to limitation about diffusion dependency and computation cost**
>
> In our response to **Reviewer 8wzm**, we analyze the computational cost of FedPrecise. Since each client performs adaptive segmentation only once, its overhead is only 5.07% compared to FedDEO and remains lower than FedBiP. We also report results on two alternative backbones beyond Stable Diffusion v1.5. The experiments show that, like other diffusion-based methods, FedPrecise is naturally influenced by backbone performance.
>
> We will further discuss limitations of these two aspects in the revised version.
>
> |backbone|method|Q|S|C|R|I|P|accuracy|
> |---|---|---:|---:|---:|---:|---:|---:|---:|
> |SD1.4|FedPrecise|68.91|81.30|85.06|86.73|59.71|71.01|75.45|
> |SD1.4|FedDEO|71.94|72.01|79.12|84.94|45.40|67.05|70.08|
> |SD1.4|FedBiP|66.06|71.76|82.95|87.86|52.15|70.50|71.88|
> |SD1.4|FedLMG|54.42|68.45|76.82|77.56|50.51|62.96|65.12|
> |SD2.1|FedPrecise|72.70|80.01|84.53|88.44|59.08|73.92|76.45|
> |SD2.1|FedBiP|73.39|69.85|79.50|88.63|58.08|70.75|73.37|
> |SD2.1|FedDEO|71.36|74.28|80.12|87.37|50.69|73.29|72.85|
> |SD2.1|FedLMG|58.76|73.69|80.14|81.06|54.74|67.50|69.31|

---

> > ### Author Rebuttal · Reviewer_b4ib · 2026-04-02
> >
> > The author provided extensive experimental results in response to my questions. I acknowledge the contributions made by this paper. Therefore, I am willing to raise my score by one point to support the author.

---

> > > ### Author Response · Authors · 2026-04-02
> > >
> > > We once again thank you for your thoughtful review and valuable feedback.
> > >
> > > Your comments gave us the opportunity to clarify the key components of our method and to present additional meaningful experimental results. We truly appreciate the time and effort you've dedicated to reviewing our paper, as well as your careful consideration throughout the review process.

---

### Official Review · Reviewer_pDfA · 2026-03-12

**Soundness:** 3
**Presentation:** 3
**Significance:** 2
**Originality:** 2
**Overall Recommendation:** 3
**Confidence:** 3

**Summary:**

The paper proposes FedPrecise, a novel framework for One-Shot Federated Learning (OSFL) to address feature space heterogeneity across clients. FedPrecise leverages a pretrained Latent Diffusion Model (LDM) on the server to synthesize client-like data. To guide the generation, clients perform a local adaptive segmentation by measuring the noise-prediction MSE and optimize stage-wise token descriptors using a MLP hypernetwork. These lightweight descriptors and segmentation boundaries are then transmitted to the server in a single communication round. The paper evaluate the method on multiple heterogeneous datasets, demonstrating its effectiveness in downstream classification accuracy and communication efficiency.

**Compliance With Llm Reviewing Policy:**

Affirmed.

**Final Justification:**

Some of the weaknesses have been addressed. However, my main concern about technological contribution are still remained.

**Key Questions For Authors:**

See the weakness.

**Limitations:**

yes

**Strengths And Weaknesses:**

**Strengths**
- **Communication Efficiency**: The proposed method achieves an extremely low communication overhead by only uploading stage-wise descriptors rather than complete model parameters.
- **Comprehensive Experimental Evaluation**: The authors conduct extensive experiments across diverse datasets, including standard OSFL benchmarks (DomainNet, PACS, OfficeHome) and challenging real-world domains (DermaMNIST, UCMerced).
- **Privacy Preservation**: By restricting the process to a one-shot communication and avoiding raw data or full model parameter uploads, the framework demonstrates strong resistance to Membership Inference Attacks (MIA) compared to FL baselines.

**Weaknesses**
1. This Paper lacks algorithmic novelty. The core contribution of FedPrecise appears to be extensions and integrations of existing techniques. For example, the OSFL paradigm via local descriptions is from FedDEO, while the multi-stage prompt control mechanism is from ProSpect. The paper lacks fundamental architectural or theoretical innovations.
2. The paper aims to solve feature space heterogeneity in OSFL. However, it fails to provide a rigorous analysis or compelling intuition for why introducing the stage-wise prompt control mechanism can address the feature space heterogeneity compared to a single prompt. The mapping between stage-wise diffusion and feature space heterogeneity mitigation remains ambiguous.
3. The paper lacks solid theoretical or empirical evidence to prove that the MSE-based Local Adaptive Segmentation leads to an optimal partitioning for capturing heterogeneous features. This segmentation maybe relies heavily on hyperparameters (e.g., the block size $m$ and $\Delta_{min}$). In Sensitivity Analyses, the block size significantly affects the performance. Furthermore, the best block size $m=10$ is the same to the setting of ProSpect, which fails to prove the contribution of Local Adaptive Segmentation.
4. The theoretical bounds presented in Theorem A.3 do not provide new insights. As acknowledged in Remark A.4, the proof is essentially a multi-step expansion of the single-stage KL-divergence upper bound already established by FedDEO.

---

> ### Author Rebuttal · Authors · 2026-03-31
>
> Thanks for the thorough review and valuable feedback!
>
> > **Response to weakness about novelty**
>
> We would like to clarify that this paper first identifies a major limitation of prior OSFL works, namely that they insufficiently address the feature-space heterogeneity in FL. Based on this weakness, the core contribution of our work is to propose, for the first time in OSFL, a method that leverages the stage-wise structure of diffusion models to address feature-space heterogeneity. This aspect has not been explored by FedDEO or other prior OSFL methods.
>
> For the specific algorithmic design, we note that ProSpect and the OSFL setting are fundamentally different: the former focuses on precise inversion and editing of attributes of a small number (1 to 4) target images, whereas the latter aims to summarize the common semantic attributes across a larger number of images on the client under a one-shot budget and guide the server to reconstruct images consistent with those attributes. To lower the client-side computation overhead, we design a lightweight MLP network to learn descriptors instead of reusing ProSpect’s heavy attention-based hypernetwork. This makes it possible to train multi-stage descriptors without significantly increasing client-side computation. At the same time, to explicitly address feature-space heterogeneity, we originally introduce adaptive segmentation to capture domain-specific data characteristics, which is also not considered in ProSpect.
>
> > **Response to weakness about stage-wise prompt control**
>
> In FedPrecise, conditioning is achieved by providing multi-stage guidance that better aligns the server-side generation process with attributes of client data. As a result, the synthesized data more faithfully preserves feature-level properties, thereby reducing the degradation of the global classifier caused by feature shift. We further report the MMD [1] and CMMD [2] between real client data and images generated by FedPrecise, FedDEO, and single-token conditioning.
>
> | Metric(×10^-3) | FedPrecise (Single-Stage) | FedDEO | FedPrecise |
> | - | -: | -: | -: |
> | MMD ↓ | 157.54 | 136.18 | 100.59 |
> | CMMD ↓ | 8.29 | 5.37 | 3.45 |
>
> Considering that FedPrecise uses only a single token per stage, compared to the 77 tokens used in FedDEO, the lower scores further demonstrate that FedPrecise can more efficiently guide the server to generate images that are closer to the semantic characteristics of client data within a more compact representation.
>
> The visualization in Figure 4 also shows that, on highly rare domains (e.g., skin lesion images), FedPrecise significantly reduces oversharpened and stylized artifacts compared to the current SOTA baseline FedBiP. These artifacts can mislead the global classifier in learning the true data distribution.
>
> > **Response to weakness about Local Adaptive Segmentation**
>
> We regret any confusion caused by our theory section. Actually, m denotes the minimum unit used to partition the 1000 diffusion timesteps. As shown in the Appendix, setting $m = 100$ provides a practical trade-off between performance and computation cost.
>
> Further analysis is provided in our response to **Reviewer b4ib**. The results show that, except for extreme cases such as $M = 10$ or $Delta_{min} = 10$ (where sampling is insufficient or the minimum segment length is smaller than the diffusion sampling interval), adaptive segmentation remains stable across different parameter choices.
>
> The stage number $S = 10$ is a prior that has been shown to work well in ProSpect, and our sensitivity analysis further justifies this choice. Moreover, ablation results show that although $S$ affects performance, under $S = 10$, adaptive segmentation remains a key factor enabling FedPrecise to outperform other baselines, demonstrating its importance and contribution.
>
> > **Response to weakness about theoretical analysis**
>
> To provide a dedicated theoretical analysis for FedPrecise, Theorem A.3 introduces a new assumption aligned with multi-stage guidance and independently derives an upper bound on the KL divergence between the generated and real data distributions under stage-wise conditioning, which is not covered by prior methods. We further observe that this bound can be expressed in an information gain form. For comparison, we rewrite the single-stage bounds in FedDEO in a similar form, showing that multi-stage guidance, as a new OSFL paradigm, is as reliable as validated single-stage approaches for guiding data synthesis.
>
> Moreover, Remark A.4 interprets the KL upper bound in terms of information gain, providing a more intuitive explanation. In practice, since multi-stage descriptors contain more non-redundant information, FedPrecise can achieve a tighter bound than FedDEO, which is also supported by our experimental results.
>
> [1] A Kernel Two-Sample Test, JMLR 2012.
>
> [2] Rethinking FID: Towards a Better Evaluation Metric for Image Generation, CVPR 2024.

---

> > ### Author Rebuttal · Reviewer_pDfA · 2026-04-03
> >
> > I would like to thank the authors for their detailed response. Some of the weaknesses have been addressed. However, my main concern about technological contribution are still remained. I intend to increase my score to 3.

---

> > > ### Author Response · Authors · 2026-04-06
> > >
> > > Thank you for your feedback and for taking the time to re-evaluate our work!
> > >
> > > However, with regard to your concern, please allow us to briefly clarify once again: The novelty of this work lies in identifying feature-space heterogeneity as an insufficiently addressed challenge, and in being the first to explicitly address it through the stage-wise condition. This problem setting and solution have not been explored in prior OSFL works.
> > >
> > > FedPrecise is also not a simple application of ProSpect. We replace its hypernetwork with a lightweight MLP and further introduce an original adaptive segmentation module. Our ablation studies show that both components are necessary for FedPrecise to outperform prior methods in the OSFL setting, which further highlights the originality of our contribution.
> > >
> > > We again thank you for taking the time to read our response, and we sincerely hope that our clarification will better address your concern.

---

### Decision · Program_Chairs · 2026-04-30

**Decision:**

Reject

**Comment:**

The paper receives the following mixed ratings: Weak reject, Accept, Weak accept, Weak reject. The authors introduce FedPrecise, a stage-precise framework to address feature space heterogeneity in one‑shot federated learning. Reviewers agree the method is well‑motivated, empirically strong across diverse domains, and communication‑efficient. However, concerns are raised on issues including limited algorithmic novelty, incomplete theoretical justification, and scalability/generalization under broader settings. After rebuttal, one reviewer moves to accept, while two maintain weak‑reject positions and one stays at weak accept. After careful consideration, the AC decided not to recommend the paper at this time. We encourage the authors to revise and resubmit to an appropriate future venue.